Corrected: Publisher correction

# Chloroquine modulates antitumor immune response by resetting tumor-associated macrophages toward M1 phenotype

Degao Chen[1], Jing Xie[1], Roland Fiskesund[1], Wenqian Dong[1], Xiaoyu Liang[1], Jiadi Lv[1], Xun Jin[1], Jinyan Liu[1], Siqi Mo[1], Tianzhen Zhang[1], Feiran Cheng[1], Yabo Zhou[1], Huafeng Zhang[2], Ke Tang[2], Jingwei Ma[2], Yuying Liu[1,3] & Bo Huang [1,2,3]

Resetting tumor-associated macrophages (TAMs) is a promising strategy to ameliorate the immunosuppressive tumor microenvironment and improve innate and adaptive antitumor immunity. Here we show that chloroquine (CQ), a proven anti-malarial drug, can function as an antitumor immune modulator that switches TAMs from M2 to tumor-killing M1 phenotype. Mechanistically, CQ increases macrophage lysosomal pH, causing $Ca^{2+}$ release via the lysosomal $Ca^{2+}$ channel mucolipin-1 (Mcoln1), which induces the activation of p38 and NF-κB, thus polarizing TAMs to M1 phenotype. In parallel, the released $Ca^{2+}$ activates transcription factor EB (TFEB), which reprograms the metabolism of TAMs from oxidative phosphorylation to glycolysis. As a result, CQ-reset macrophages ameliorate tumor immune microenvironment by decreasing immunosuppressive infiltration of myeloid-derived suppressor cells and Treg cells, thus enhancing antitumor T-cell immunity. These data illuminate a previously unrecognized antitumor mechanism of CQ, suggesting a potential new macrophage-based tumor immunotherapeutic modality.

[1] Department of Immunology and National Key Laboratory of Medical Molecular Biology Institute of Basic Medical Sciences, Chinese Academy of Medical Sciences, Beijing 100005, China. [2] Department of Biochemistry and Molecular Biology, Tongji Medical College Huazhong University of Science and Technology, Wuhan 430030, China. [3] Clinical Immunology Center, Chinese Academy of Medical Sciences, Beijing 100005, China. Degao Chen, Jing Xie and Roland Fiskesund these authors contributed equally to this work. Correspondence and requests for materials should be addressed to B.H. (email: tjhuangbo@hotmail.com)

Understanding the interaction between tumor cells and immune cells is pivotal to designing and developing novel immunotherapeutic against cancers. To date, studies have mainly concentrated on macrophages, because they are strikingly accumulated in tumor microenvironment, as evidenced not only by mouse tumor models but also from patient samples[1,2]. As the major tumor-infiltrating immune cell population, these tumor-associated macrophages (TAMs) are commonly educated by tumor cells to become their partners in crime, promoting tumor immune escape, angiogenesis, tumor growth, and metastasis. Therefore, targeting TAMs is considered as a promising strategy in cancer immunotherapy[3–5]. Notwithstanding their tumor-promoting effects, macrophages are actually capable of killing tumor cells by releasing nitrogen oxide (NO) and interferon-γ (IFN-γ)[6,7]. Notably, TAMs are phenotypically described as M2 macrophages that are alternatively activated by Th2 cytokines interleukin (IL)-4, IL-13, and other factors. By contrast, tumor-killing macrophages are typically described as M1 macrophages that are classically activated by Th1 cytokines such as IFN-γ[8–10]. Therefore, an ideal approach to target tumor-infiltrating macrophages is not through depleting them but rather converting M2 TAMs into M1 antitumor macrophages.

As professional phagocytes, macrophages are highly capable of taking up extracellular materials and effectively degrading them in lysosomes. This degrading process strictly relies on the acidic lysosomal pH[11,12]. Therefore, modifying lysosomal pH value undoubtedly influences the fundamental phagocytosis function of macrophages. A fundamental property of M2 macrophages is

their use of phagocytosis to repair damaged tissues[8–10]. By contrast, M1 macrophages release proinflammatory cytokines to promote inflammation and exacerbate tissue damage[8–10]. Therefore, altering lysosomal pH might be a potential strategy to reset the phenotype and function of macrophages. Several alkaline agents including chloroquine (CQ) are known to be trapped in lysosomal compartments, leading to the increased lysosomal pH value[13]. CQ is a weak base that has been widely used in the clinic to treat malaria[14]. Intriguingly, recent studies have highlighted that CQ is a promising antitumor agent. Mechanistically, its antitumor effect has been ascribed to direct targeting of tumor cells and/or stromal endothelial cells[15,16]. However, whether CQ employs a macrophage-modifying strategy against cancer remains unexplored. In the present study, we provide evidence that CQ functions as an immune modulator and mediates its antitumor efficacy via resetting TAMs from M2 to M1 phenotype.

## Results

**CQ-mediated antitumor effect is T-cell dependent**. CQ, a clinically used antimalarial drug, has shown promising antitumor function in clinical trials for late-stage cancers[17]. Previous reports have indicated that 50 mg kg$^{-1}$ CQ administration results in 3–13 μM blood concentration[18,19]. Therefore, in this study, we used 75 mg kg$^{-1}$ and 10 μM CQ for in vivo and in vitro studies, respectively. Using a B16 melanoma-bearing mouse model (~ 60 mm$^3$ tumor size), we confirmed that intraperitoneal injection of CQ (75 mg kg$^{-1}$) effectively inhibited melanoma growth and prolonged the survival of the mice (Fig. 1a, b). In addition, in

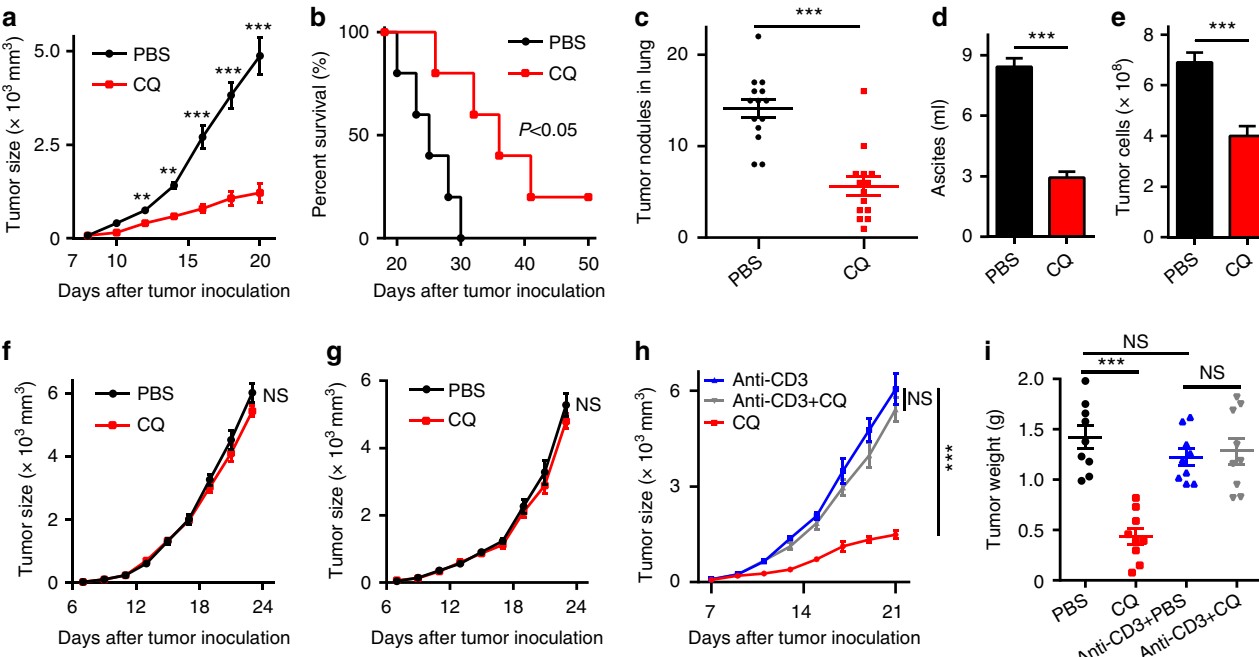

**Fig. 1** CQ-mediated antitumor effect is T-cell dependent. **a** C57BL/6 mice ($n = 6$) with subcutaneous B16 melanoma were i.p. treated with PBS or 75 mg kg$^{-1}$ CQ and the tumor growth was monitored for 2 weeks. **b** C57BL/6 mice ($n = 10$) with subcutaneous B16 melanoma were i.p. treated with PBS or 75 mg kg$^{-1}$ CQ for 2 weeks. The long-term survival was recorded according to Kaplan–Meier analysis and analyzed by log-rank (Mantel–Cox) test. **c** C57BL/6 mice ($n = 14$) were intravenously injected with 5 × 10$^4$ B16 cells. One week later, the mice were peritoneally treated with PBS or 75 mg kg$^{-1}$ CQ for 2 weeks and then killed for lung tumor nodules determination. **d, e** BALB/c mice ($n = 6$) with H22 hepatocarcinoma ascites were i.p. treated with PBS or 75 mg kg$^{-1}$ CQ for 1 week and then killed. The tumor ascites volume **d** and the tumor cell number **e** were measured. **f** Nude mice ($n = 12$) with B16 melanoma were received daily i.p. injection of PBS or 75 mg kg$^{-1}$ CQ for 2 weeks. Tumor growth was measured. **g** Nude mice ($n = 10$) with subcutaneous H22 hepatocarcinoma were i.p. injected with PBS or 75 mg kg$^{-1}$ CQ for 2 weeks. Tumor growth was followed. **h** C57BL/6 mice ($n = 9$) with B16 melanoma were treated with anti-CD3 neutralizing antibody and/ or CQ for 2 weeks. Tumor growth was monitored. **i** BALB/c mice ($n = 6$) with H22 muscle hepatocarcinoma were intratumorally injected with PBS or CQ with or without anti-CD3 neutralizing antibody treatment. After 1 week, the mice were killed and the tumor weight was measured by the weight of right thigh minus the left. Data shown are representative of three independent experiments and error bars represent mean ± SEM. *$P < 0.05$; **$P < 0.01$; ***$P < 0.001$; NS, not statistically significant (Student's $t$-test)

the B16 lung metastasis model, CQ treatment remarkably decreased the number of tumor nodules in the lungs (Fig. 1c and Supplementary Fig. 1a). Furthermore, in the H22 hepatocarcinoma malignant ascites model, intraperitoneal injection of CQ significantly ($P < 0.001$) reduced the volume of ascites as well as the number of tumor cells (Fig. 1d, e). Thus, we confirm that CQ possesses intrinsic antitumor activities. CQ's antitumor activity was initially ascribed to its inhibitory effect on the autophagy of cancer cells. Recent studies have also indicated that CQ can target stromal endothelial cells, inducing tumor vessel normalization[16]. However, whether CQ treatment mobilizes other tumor-fighting mechanisms utilizing the host immune system for its antitumor effect remains unclear. Intriguingly, in T-cell-deficient nude mice, we found that the tumor-inhibiting effect of CQ against B16 melanoma and H22 hepatocarcinoma was lost (Fig. 1f, g). This was further confirmed through T-cell depletion in C57BL/6 and BALB/c mice (Fig. 1h, i and Supplementary Fig. 1b). Together, these data suggest that the antitumor effect of CQ appears to be ascribed to its mobilizing T-cell immunity.

**CQ-induced antitumor T-cell immunity is macrophage dependent**. Next, we studied the mechanisms behind the above described CQ-mediated antitumor T-cell immunity. When splenic T cells were stimulated with anti-CD3 and CD28 antibodies in the presence or absence of CQ, we found that CQ ($5 \mu M$) had no effect on T-cell proliferation, and that a high CQ concentration ($10 \mu M$) even resulted in decreased T-cell proliferation (Supplementary Fig. 2a). We also isolated splenic T cells from CQ ($75$ $mg kg^{-1}$) and control treated mice, respectively. Consistent with the proliferation experiment above, T cells from CQ-treated mice where phenotypically indistinguishable from those of sham-treated mice (Supplementary Fig. 2b) with regard to apoptosis, T-cell activation (CD69 and CD137), lysosome status (Lamp1), as well as T-cell anergy (PD1 and CTLA-4). Based on this result, we hypothesized that CQ dose not act directly on T cells for promoting T-cell antitumor immunity. Dendritic cells (DCs), as professional antigen-presenting cells, have critical roles in T-cell immunity. When bone marrow-derived DCs (BMDCs) were incubated with CQ in the presence or absence of lipopolysaccharide (LPS), CQ did not have any effect on DC maturation, evaluated by the unchanged expression of CD80, CD86, MHC-II, and CCR7 (Supplementary Fig. 2c). Given that tumor cells are the major cell population in a given tumor, we also co-cultured CQ-treated B16 or H22 tumor cells with DCs and the expression of the above-mentioned molecules was also unaltered (Supplementary Fig. 2d), suggesting that CQ also does not directly target DCs to promote antitumor T-cell immunity. Unlike DCs, TAMs are widely distributed in tumor microenvironment, where they are persuaded to not only promote tumor angiogenesis and metastasis but also suppress antitumor immune response. Notwithstanding such tumor-promoting effect, macrophages with M1 phenotype are actually intrinsically capable of killing tumor cells. To determine whether macrophages are involved in CQ-mediated antitumor effect, we treated mice with clodronate liposomes[20,21]. As expected, macrophages were effectively depleted, whereas T cells were not influenced (Supplementary Fig. 3a,b). We found that macrophage depletion disrupted the inhibitory effect of CQ on B16 melanoma, as well as H22 hepatocarcinoma growth (Fig. 2a–c). IFN-γ is a critical effector molecule in T-cell-mediated antitumor immunity[22]. In B16 melanoma cells and in H22 tumor ascites, CQ treatment boosted the production of IFN-γ by tumor-infiltrating CD8+ T cells and macrophage depletion effectively nullified this (Fig. 2d, e). To further demonstrate that CQ treatment enhances IFN-γ production by CD8+ T cells, we depleted CD8+ T cells in tumor-bearing mice. We found that

CD8+ T-cell depletion resulted in the loss of most of the IFN-γ in tumor tissue (Fig. 2f). In addition to IFN-γ, the expression of PD1 and CTLA-4, two critical checkpoint molecules that negatively regulate T-cell immune response[23,24], were significantly decreased ($P < 0.05$ in melanoma; $P < 0.01$ in hepatocarcinoma) in tumor-infiltrating CD8+ T cells by CQ monotreatment but not when CQ was combined with clodronate (Fig. 2g, h). T-bet is an essential transcription factor for IFN-γ production in CD8+ T cells[25,26]. We found that T-bet was upregulated in tumor-infiltrating CD3+CD8+ T cells upon CQ treatment, whereas the expression of PD1/CTLA-4 was downregulated in CD3+CD8+T-bet+ T cells (Supplementary Fig. 3c), consistent with previous report that T-bet represses PD-1 expression[27]. To further clarify the role of T-cell immunity in CQ-mediated antitumor effect, we adoptively transferred CD45.2+ OT-I T cells to OVA-B16 melanoma-bearing CD45.1+ C57BL/6 mice treated with or without CQ, and found that CQ treatment enhanced the tumor growth inhibition mediated by the adoptively transferred CD8+ T cells, which however was blocked by concomitant macrophage depletion (Fig. 2i). Consistently, analysis by flow cytometry and immunohistochemical staining showed that tumor-infiltrating OT-I T cells upregulated IFN-γ and granzyme B within the tumor tissues (Supplementary Fig. 3d, e). In addition, OT-I T cells isolated from CQ-treated mice were more efficient in killing OVA-B16 cells ex vivo, compared with OT-I T cells from untreated mice (Supplementary Fig. 3f). Thus, CQ potentially mobilizes macrophages to activate T cells as the main effector arm against tumors. Together, these findings suggest that CQ initiates a macrophage-dependent pathway that enhances antitumor T-cell immunity.

**CQ resets tumor-associated M2 macrophages to M1 phenotype**. Next, we investigated how CQ regulates macrophages, leading to tumor suppression. Classical activation of macrophages (M1 phenotype) leads to the upregulation of costimulatory molecules and confers them the ability to kill tumor cells by producing NO and IFN-γ or via other ways[8–10]. We found that both bone marrow-derived and peritoneal macrophages upregulated the expression of antitumor inflammatory cytokines IFN-γ, IL-12p40, and tumor necrosis factor-α (TNF-α) after CQ treatment, concomitant with the upregulation of costimulatory molecules CD80, CD86, and MHC-II (Supplementary Fig. 4a, b). Meanwhile, CQ also promoted the expression of the above-mentioned cytokines and costimulatory molecules, as well as NO production in IL-4-conditioned M2 macrophages (Fig. 3a–c), and IFN-γ, IL-12p70, and TNF-α were abundantly present in the supernatants (Supplementary Fig. 4c). Consistently, when we used CQ to treat IL-4-conditioned M2 macrophages, the number of CD206+CD301+ M2 macrophages[28] decreased sharply (Fig. 3d). In addition, upon CQ treatment, the expression of inducible nitric oxide synthase (iNOS, a M1 marker) was remarkably upregulated and the expression of arginase 1 (Arg1, a M2 marker) as well as immunosuppressive factors IL-10, transforming growth factor (TGF)-β1, and IDO1 were downregulated in IL-4-conditioned bone marrow or Raw264.7 macrophages (Fig. 3e and Supplementary Fig. 4d, e). Intriguingly, CQ-treated M2 macrophages were slightly able to induce tumor cell death and inhibition of NO production resulted in the disruption of such slight killing (Supplementary Fig. 4f), suggesting that CQ-polarized macrophages may directly induce a small-scale tumor cell death in a NO-dependent manner. In addition, although most CQ-treated macrophages belonged to IFN-γ/IL-12-negative phenotype, CQ treatment resulted in the loss of Arg1 expression in 30% IL-12−IFN-γ− macrophages (Fig. 3f). On the other hand, when we co-cultured CQ-treated B16 or H22 tumor cells with macrophages,

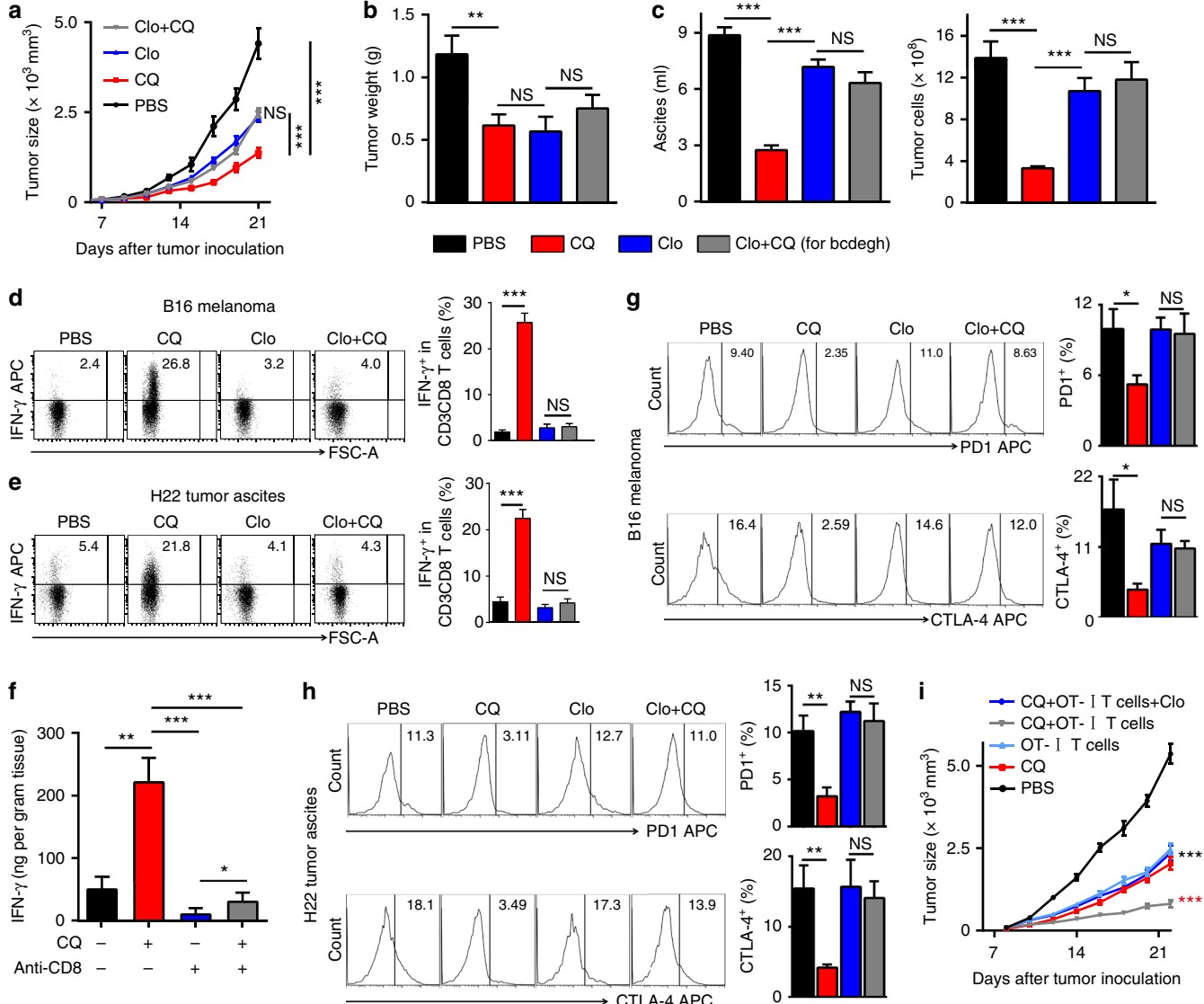

**Fig. 2** CQ-induced antitumor T-cell immunity is macrophage dependent. **a** C57BL/6 mice ($n = 8$) bearing subcutaneous B16 melanoma with or without clodronate liposomes treatment were intratumorally injected with PBS or CQ for 2 weeks (Clo: clodronate liposomes). The tumor growth was followed. **b** BALB/c mice ($n = 6$) bearing H22 muscle hepatocarcinoma with or without clodronate liposomes treatment received intratumoral injections of PBS or CQ (Clo: clodronate liposomes). After 1 week of treatment, the mice were killed and the tumor weight was measured by the weight of right thigh minus the left. **c** BALB/c mice ($n = 6$) bearing H22 hepatocarcinoma ascites with or without clodronate liposomes treatment received i.p. injections with PBS or CQ. After 1 week, the mice were killed to measure the tumor ascites volume (left) and the tumor cell number (right). **d, e** Tumor-bearing mice with or without clodronate liposomes treatment were i.p. injected with PBS or CQ. IFN-γ expression in CD3+CD8+ tumor-infiltrating lymphocytes (TILs) from B16 melanoma **d** ($n = 6$) or H22 hepatocarcinoma ascites **e** ($n = 6$) was analyzed by flow cytometry. **f** B16 melanoma mice ($n = 5$) with or without anti-CD8 neutralizing antibody treatment were i.p. injected with PBS or CQ for 1 week. IFN-γ in tumor tissue was measured by ELISA. **g, h** Tumor-bearing mice with or without clodronate liposomes treatment received i.p. injections with PBS or CQ. Representative flow cytometric analysis and quantification of PD-1 and CTLA-4 in CD3+CD8+ TILs from subcutaneous B16 melanoma **g** ($n = 6$) and H22 hepatocarcinoma ascites **h** ($n = 6$). **i** OVA-B16 melanoma-bearing CD45.1+ C57BL/6 mice ($n = 10$) were treated with or without CQ, concomitant with adoptive transfer of OVA-specific CD8+ T cells and/or clodronate liposomes treatment. Tumor growth was followed. Black asterisk, PBS compared with CQ, OT-I T cells, or CQ + OT-I T cells + Clo; red asterisk, CQ + OT-I T cells compared with CQ or CQ + OT-I T cells + Clo. Data shown are representative of three independent experiments and error bars represent mean ± SEM. *$P < 0.05$; **$P < 0.01$; ***$P < 0.001$; NS, not statistically significant (Student's $t$-test)

we found that CQ-treated tumor cells had minor effect on CD80 and CD86 expression, slightly upregulated MHC-II expression, but strikingly upregulated Arg1 and IL-10 expression (Supplementary Fig. 4g), suggesting that CQ-effected tumor cells do not contribute to M1 macrophage polarization. We then validated the results in vivo. In the H22 hepatocarcinoma ascites model, CQ treatment caused iNOS increase and Arg1 decrease in malignant ascites macrophages (Fig. 3g). In the B16 melanoma

subcutaneous tumor model, CQ treatment consistently upregulated iNOS, while downregulating Arg1 in melanoma-infiltrating macrophages (Fig. 3h), concomitant with the increased expression of IFN-γ, IL-12p40, IL-12p35, and TNF-α, and downregulation of IL-10 in tumor tissues (Fig. 3i). To further prove the resetting effect of CQ on M2 macrophages, we additionally tested their immunosuppressive function. Supernatants derived from IL-4-conditioned M2 macrophages significantly inhibited the

proliferation of CD8$^+$ and CD4$^+$ T cells in an OVA-mediated T-cell assay. However, this T-cell suppression was relieved once the M2 macrophages were treated with CQ (Fig. 3j, k). In addition to murine macrophages, we also analyzed human macrophages. IL-4-conditioned human M2 macrophages were also transited to M1 phenotype by CQ treatment, as evidenced by the upregulation of IFN-γ, IL-12p40, IL-12p70, and TNF-α, and concomitant downregulation of CD163, CD206$^+$CD64$^-$, TGF-β1, IL-10, and Arg1 (Supplementary Fig. 4h-j). IPI-549, an inhibitor of PI3Kγ, is known as a M1 polarizer[3]. Here we also combined it with CQ to treat B16 melanoma and H22 hepatocarcinoma. We found that

the inhibitory effect of CQ on the growth of B16 melanoma and H22 hepatocarcinoma was enhanced by IPI-549, along with further depression of T-cell PD1 and CTLA-4 expression (Supplementary Fig. 4k-m), suggesting that PI3Kγ inhibition has a synergistic effect on CQ-triggered antitumor immunity. Together, these results indicate that CQ is capable of resetting macrophages from a tumor-promoting to a tumor-inhibiting phenotype.

**M1 macrophages are induced by CQ activated p38 and NF-κB.** Next, we investigated the molecular mechanism through which

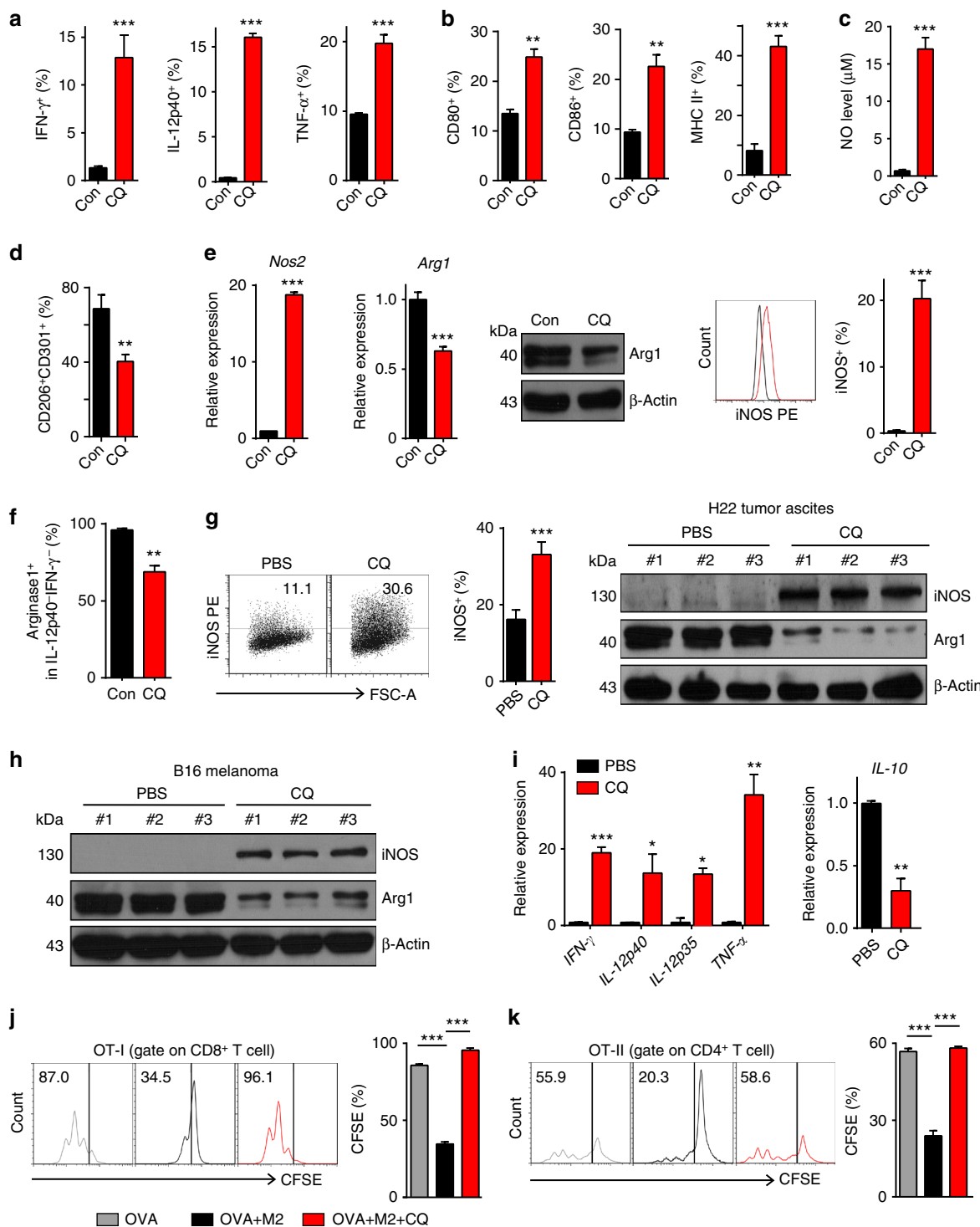

CQ resets TAMs. To this end, we focused on mitogen-activated protein kinase (MAPK) and nuclear factor-κB (NF-κB) signal molecules, because (1) MAPK and NF-κB are activated in IFN-γ/LPS-induced M1 macrophages[8,10,29–31]; (2) MAPK and NF-κB are the key regulator of proinflammatory factors[32]; and (3) pro-inflammatory factors are upregulated in the above settings. By analyzing MAPK three members Erk1/2, JNK, and p38, we found that only p38 but not Erk or JNK was phosphorylated in M2 macrophages and Raw264.7 cells after CQ treatment (Fig. 4a and Supplementary Fig. 5a, b), implying that p38 might be involved in CQ-induced macrophage phenotype switch. SB203580, an inhibitor of p38, was capable of blocking LPS/IFN-γ-polarized M1 macrophages (Supplementary Fig. 5c, d). Consistently, SB203580 blocked CQ's effect on primary macrophages and Raw264.7 cells, as evidenced by the unchanged expression of Arg1 and iNOS (Fig. 4b, c and Supplementary Fig. 5e). Moreover, knockout of p38 with CRISPR/Cas9 technology resulted in the loss of the ability of CQ to reset macrophage phenotype (Fig. 4d, e and Supplementary Fig. 5f-h), suggesting that CQ indeed utilizes p38 signaling for M1 phenotype development. Then, we tested NF-κB by checking its nuclear translocation. We found that CQ treatment resulted in the entry of NF-κB p65 into the nucleus of IL-4-conditioned bone marrow and Raw264.7 macrophages (Fig. 4f and Supplementary Fig. 5i). In particular, in Raw264.7 macrophages, 90–95% cells showed NF-κB in the nucleus after CQ treatment; however, untreated cells had almost no NF-κB in the nucleus (Supplementary Fig. 5i). On the other hand, blocking this p65 entry process with NF-κB inhibitor JSH-23 resulted in the loss of ability of CQ to reset M2 to M1 macrophages (Fig. 4g and Supplementary Fig. 5j). In addition, we found that the expression of iNOS was downregulated in M1 macrophages after the inhibition of the entry of NF-κB p65 into the nucleus (Fig. 4h). Taken together, these data suggest that the activation of MAPK p38 and NF-κB is required for CQ-mediated M1 macrophage development.

**Lysosomal Ca$^{2+}$ release activates p38 and NF-κB**. Next, we investigated the signaling pathway through which CQ induces the activation of p38 and NF-κB. Being a weak base, CQ is trapped upon protonation in acidic compartments such as lysosomes[33]. Notably, it has been reported that M2 macrophage lysosomes are more acidic than M1 lysosomes[34]. Staining lysosomes with LysoSensor Green, we found that M2 macrophages had more acidic lysosomes than M1 macrophages (Fig. 5a). Moreover, measurement of lysosomal pH value showed that CQ treatment increased M2 lysosomal pH from 4.53 to 5.38, which was similar to the lysosomal pH value (pH 5.30) of LPS/IFN-γ-polarized M1 macrophages (Fig. 5b and Supplementary Fig. 6a). Besides CQ, other agents are also able to increase lysosomal pH value. Here we

tested bafilomycin A1 (Baf). Although Baf could increase the lysosomal pH in IL-4-conditioned M2 macrophages from 4.56 to 5.09 (Supplementary Fig. 6b), it was less able to increase iNOS, IFN-γ, and IL-12p40 expression, compared with CQ (Supplementary Fig. 6c). Furthermore, when we knocked down Atp6v1c1, a subunit of V-ATPase, to raise lysosomal pH (Supplementary Fig. 6d, e), we observed only a weak upregulation in the expression of iNOS, IFN-γ, and IL-12p40 (Supplementary Fig. 6f), suggesting that CQ has other unique feature(s) besides altering lysosomal pH value. In line with the increased pH value, the number of lysosomes was observed to be elevated in CQ-treated M2 macrophages (Fig. 5c). Consistently, lysosomal markers Lamp1 and Lamp2 were remarkably displayed after CQ treatment (Fig. 5d). Transcription factor EB (TFEB) is known to be critical for lysosomal biogenesis upon its translocation into the nucleus[35,36]. In line with the increased lysosome quantity, TFEB expression and its entry into the nucleus were both found to be increased upon CQ treatment in IL-4-conditioned bone marrow or Raw264.7 macrophages, as evidenced by real-time quantitative PCR (qPCR) and immunofluorescent staining (Fig. 5e, f), which was further confirmed by western blot analysis of cytoplasmic and nuclear fractionation (Supplementary Fig. 6g), supporting the above observed lysosomal alteration. Moreover, TFEB knock-down resulted in the abrogation of CQ-increased lysosome number, concomitant with the reduction of Lamp1 and Lamp2 expression (Supplementary Fig. 6h-j). However, TFEB knock-down did not affect the increased lysosomal pH value in macrophages seen after CQ treatment (Supplementary Fig. 6k). Given that the nuclear translocation of TFEB is regulated by lysosome-released calcium[37], we additionally analyzed intracellular Ca$^{2+}$. Along with increased lysosomal pH, the intracellular calcium concentration was also elevated in CQ-treated M2 macrophages (Fig. 5g). Intriguingly, using ryanodine or CGP37157 to block calcium release from the endoplasmic reticulum or mitochondria[38,39] did not affect the CQ-increased calcium levels in macrophages (Fig. 5h), suggesting that released lysosomal Ca$^{2+}$ mediates the above described increase of cytoplasmic calcium. Several Ca$^{2+}$ channels have been known to mediate lysosomal calcium release. Among them, mucolipin-1 (Mcoln1), a major lysosomal channel[40,41], was found to be upregulated by CQ (Fig. 5i). However, CRISPR/Cas9 knockout of Mcoln1 (Supplementary Fig. 7a) blocked the resetting of IL-4-conditioned M2 macrophages to M1 by CQ (Fig. 5j, k and Supplementary Fig. 7b, c). In addition, treatment with cyclosporin A (CsA), a Ca$^{2+}$ signaling inhibitor[37], also blocked the effect of CQ on macrophage phenotype switch (Fig. 5l). Moreover, p38 phosphorylation (Fig. 5j,m and Supplementary Fig. 7b) and NF-κB translocation (Fig. 5n,o and Supplementary Fig. 7d-g) were abolished by either Mcoln1 knockout or CsA treatment, as evidenced by western

**Fig. 3** CQ resets tumor-associated M2 macrophages to M1 phenotype. **a**, **b** BMDM-M2 cells were treated with or without 10 μM CQ for 24 h. The expression of IFN-γ, IL-12p40, and TNF-α was measured by flow cytometry **a** ($n = 3$). Quantification of CD80, CD86, and MHC-II in BMDM-M2 cells with or without CQ treatment **b** ($n = 3$). **c** NO production in BMDM-M2 cells lysate with or without CQ treatment were measured ($n = 3$). **d** Quantification of the number of CD206$^+$CD301$^+$ cells in BMDM-M2 cells with or without CQ treatment ($n = 3$). **e** The mRNA expression of NOS2 and Arg1 in BMDM-M2 cells with or without CQ treatment was analyzed by qPCR (left); the expression of Arg1 was analyzed by western blotting (center); the expression of iNOS was analyzed by flow cytometry (right). **f** Arginase1$^+$ cells in IL-12p40$^-$IFN-γ$^-$ M2 macrophages with or without CQ treatment were analyzed by flow cytometry ($n = 3$). **g** Representative flow cytometric analysis and quantification of iNOS in F4/80$^+$ ascites fluid macrophages, isolated from mice with H22 hepatocarcinoma that had received PBS or CQ treatment ($n = 5$) (left); the protein expression of iNOS and Arg1 in the same F4/80$^+$ ascites macrophages were also analyzed by western blotting (right) ($n = 3$). **h** Western blot analysis of iNOS and Arg1 in tumor-infiltrating macrophages after PBS or CQ treatment in mice bearing subcutaneous B16 melanoma ($n = 3$). **i** The mRNA expression of IFN-γ, IL-12p40, IL-12p35, TNF-α, and IL-10 was analyzed by real-time qPCR in B16 tumor tissues in mice with or without CQ treatment ($n = 3$). **j**, **k** Splenocytes from OT-I and OT-II TCR transgenic mice were labeled with carboxyfluorescein succinimidyl ester (CFSE) and then stimulated with OVA257-264 (**j**) and OVA323-339 (**k**) respectively in the presence of conditioned medium from PBS or CQ treated BMDM-M2 cells. Representative histograms of CD8$^+$ or CD4$^+$ T-cell proliferation (left panel) and quantification of CD8$^+$ or CD4$^+$ T-cell proliferation (right panel) were analyzed by flow cytometry after 72 h ($n = 3$). Data shown are representative of three independent experiments and error bars represent mean ± SEM. *$P < 0.05$; **$P < 0.01$; ***$P < 0.001$ (Student's $t$-test)

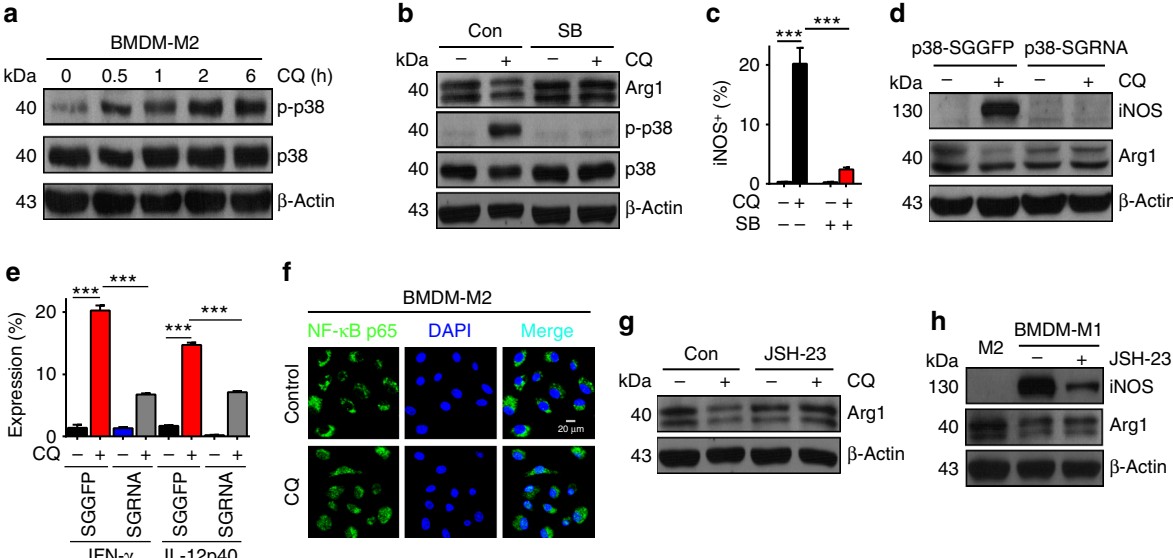

**Fig. 4** M1 macrophages are induced by CQ-activated p38 and NF-κB. **a** The expression of p-p38, total p38, and β-actin in BMDM-M2 cells treated with CQ at indicated time points were analyzed by western blotting. **b**, **c** The expression of Arg1, p-p38, total p38, and β-actin was quantified by western blotting (**b**) and the expression of iNOS was analyzed by flow cytometry (**c**) in BMDM-M2 cells that were pretreated with 10 μM SB203580 or not for 2 h before addition of CQ or PBS (*n* = 3). **d** The expression of iNOS, Arg1, and β-actin in p38-SGGFP and p38-SGRNA Raw264.7 cells with or without CQ treatment were analyzed by western blotting (*n* = 3). **e** Quantification of IFN-γ and IL-12p40 in p38-SGGFP and p38-SGRNA Raw264.7 cells with or without CQ treatment were analyzed by flow cytometry (*n* = 3). **f** NF-κBp65 in BMDM-M2 cells with or without CQ treatment was examined by confocal fluorescent microscope. Green, NF-κBp65; blue, DAPI. Scale bar, 20 μm. **g** The expression of Arg1 and β-actin in BMDM-M2 cells that were pretreated with 10 μM JSH-23 or not for 2 h before addition of PBS or CQ were analyzed by western blotting. **h** The expression of iNOS, Arg1, and β-actin were analyzed by western blotting in BMDM-M2 and BMDM-M1 cells that had been pretreated with 10 μM JSH-23 or not for 2 h. Data shown are representative of three independent experiments and error bars represent mean ± SEM. *$P < 0.05$; **$P < 0.01$; ***$P < 0.001$ (Student's *t*-test)

blotting, immunofluorescent staining, as well as cytoplasmic and nuclear fractionation analysis. Together, these data suggest that Mcoln1-mediated lysosomal calcium release induces the activation of p38 and NF-κB for M1 macrophage polarization.

**Ca²⁺-activated TFEB reprograms macrophage metabolic mode.** Macrophage M1 polarization not only causes phenotypical alterations (surface markers, cytokines, and enzymes) but also reprograms its metabolic mode. It is known that M1 macrophages are biased toward glycolysis, whereas M2 macrophages use oxidative phosphorylation[28,42,43]. In line with this notion, we found that CQ treatment increased the basal extracellular acidification rate (ECAR) in IL-4-conditioned M2 macrophages, which could be further enhanced by the addition of oligomycin that inhibits mitochondrial oxidative phosphorylation (Fig. 6a and Supplementary Fig. 8a). Meanwhile, CQ treatment significantly decreased the basal and max oxygen consumption rate (OCR) (Fig. 6b and Supplementary Fig. 8b), suggesting that CQ is capable of promoting anaerobic glycolysis in macrophages. Next, we tested whether the above described CQ-induced glucose metabolic alterations were mediated by lysosomal calcium release. We found that treating macrophages with either Mcoln1 knockout or CsA resulted in the blockade of CQ-induced metabolic alterations (Fig. 6c,d and Supplementary Fig. 8c). Moreover, we confirmed that such metabolic alterations were regulated by lysosomal calcium-activated TFEB, because TFEB knockdown blocked the CQ-induced metabolic alteration (Fig. 6e and Supplementary Fig. 9a). Given that TFEB is an important transcription factor[37,44], we bioinformatically analyzed TFEB-targeted genes. Intriguingly, many genes involved in the above-mentioned metabolic alterations were found to be regulated by TFEB (JASPAR database), which was further confirmed by real-time qPCR, especially the glucose transporters (SLC2A1 and SLC2A4) and

pyruvate kinase (PFKM and PKLR) were highly upregulated by CQ, but this upregulation was blocked after TFEB knockdown (Fig. 6f and Supplementary Fig. 9b, c). Using chromatin immunoprecipitation–qPCR (ChIP–qPCR), we also confirmed that TFEB directly bound to the promoter of these genes (Supplementary Fig. 9d). In addition, TFEB, as a member of the basic helix-loop-helix leucine-zipper family of transcription factors (TFs), directly binds to CLEAR (coordinated lysosomal expression and regulation) element in the promoters of lysosomal genes[45]. We found that CQ treatment upregulated the expression of the CLEAR gene network, including genes *Arsa*, *Arsb*, *Atp6v0e1*, *Clcn7*, *Ctsa*, *Ctsb*, *Ctsd*, *Ctsf*, *Galns*, *Gba*, *Gla*, *Gns*, *Hexa*, *Lamp1*, *Lamp2*, *Naglu*, *Neu1*, *Psap*, *Scpep1*, *Sgsh*, *Tmem55b*, and *Tpp1* in IL-4-conditioned macrophages (Supplementary Fig. 9e). Here we further clarified whether the TFEB-regulated metabolic trait is coupled with or separated from the phenotype in CQ-polarized M1 macrophages. We found that TFEB knockdown did not alter the phenotype of CQ-treated Raw264.7 macrophages, as evidenced by the unchanged expression of iNOS and Arg1 (Fig. 6g). On the other hand, when we used inhibitors to block the activities of p38 and NF-κB, CQ-mediated phenotype alteration was disrupted, whereas the CQ-mediated metabolic alterations were not influenced (Fig. 6h, i). Thus, TFEB-regulated metabolic trait is separated from the phenotype in CQ-polarized M1 macrophages. Collectively, these data suggest that CQ profoundly regulates the metabolic mode of macrophages via a cytosolic calcium-TFEB pathway, facilitating the exertion of their antitumor effect.

**CQ-reset macrophages ameliorate tumor immunosuppression.** Myeloid-derived suppressor cells (MDSCs) and regulatory T (Treg) cells are known as two major immunosuppressive cell types in tumor microenvironment that strongly suppress

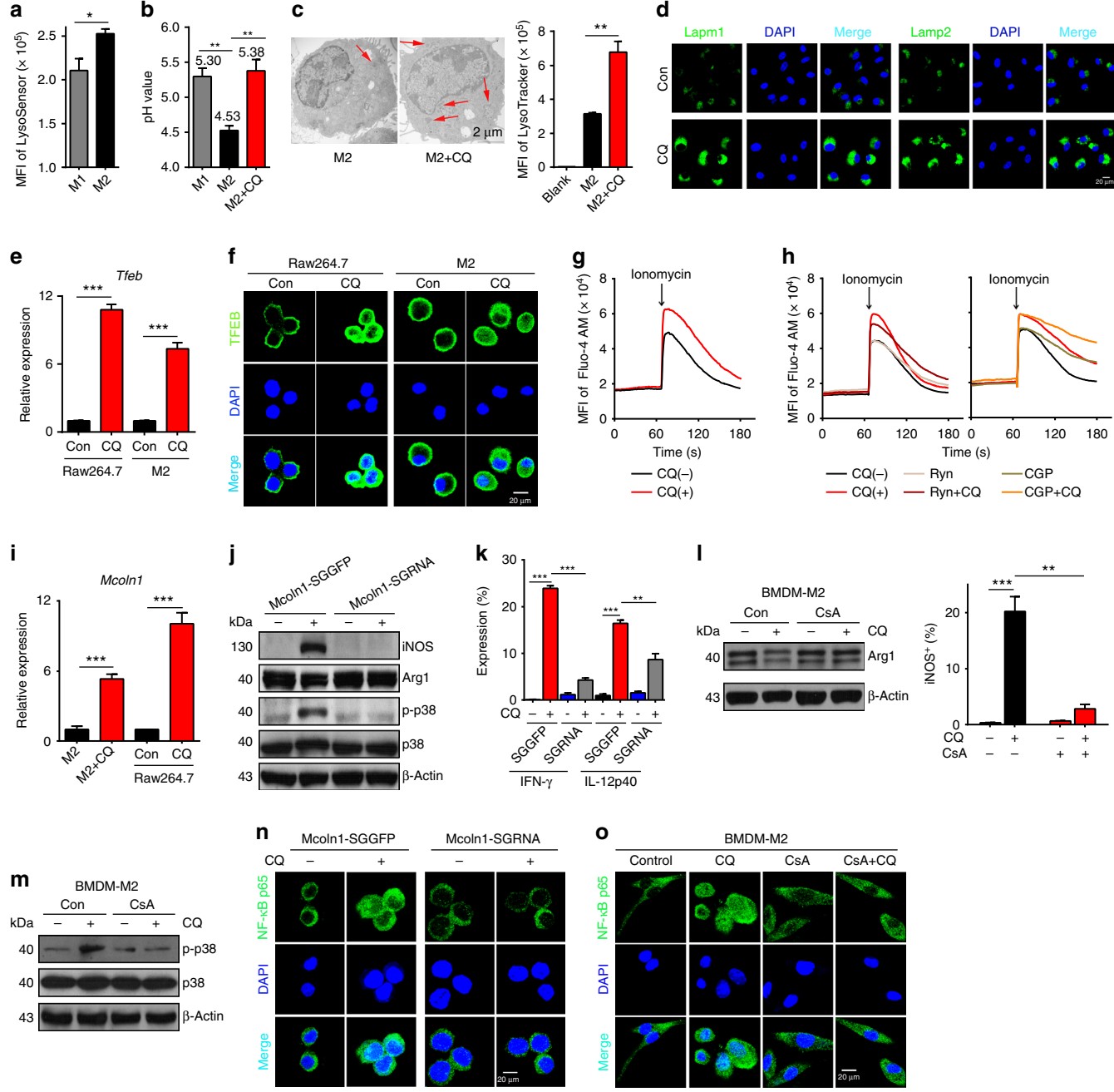

**Fig. 5** Lysosomal $Ca^{2+}$ release activates p38 and NF-κB. **a** The mean fluorescence intensity (MFI) as lysosomal pH in BMDM-M1 and BMDM-M2 cells ($n = 3$). **b** The lysosomal pH value of BMDM-M1, BMDM-M2, and CQ-treated BMDM-M2 cells ($n = 3$). **c** Left: representative picture of lysosomes (red arrow) in BMDM-M2 and CQ-treated BMDM-M2 cells. Scale bar, 2 μm; right: lysosome mass in BMDM-M2 and CQ-treated BMDM-M2 cells ($n = 3$). **d** Immunofluorescent analysis of Lamp1 and Lamp2 in BMDM-M2 and CQ-treated BMDM-M2 cells. Green, Lamp1, or Lamp2; blue, DAPI; scale bar, 20 μm. **e** The mRNA expression of *Tfeb* in Raw264.7 cells and BMDM-M2 cells with or without CQ treatment ($n = 3$). **f** Immunofluorescent analysis of TFEB in Raw264.7 cells and BMDM-M2 cells with or without CQ treatment. Green, TFEB; blue, DAPI; scale bar, 20 μm. **g** The MFI of Fluo-4 AM in BMDM-M2 cells treated with or without CQ. **h** The Fluo-4 AM MFI in BMDM-M2 cells that were pretreated for 2 h with either 100 μM Ryn, 10 μM CGP, or not, before addition of CQ or PBS. **i** The mRNA expression of *Mcoln1* in BMDM-M2 and Raw264.7 cells ($n = 3$). **j** The expression of iNOS, Arg1, p-p38, total p38, and β-actin in Mcoln1-SGGFP and Mcoln1-SGRNA Raw264.7 cells with or without CQ treatment. **k** Flow cytometric quantification of IFN-γ and IL-12p40 in Mcoln1-SGGFP and Mcoln1-SGRNA Raw264.7 cells. **l** The expression of iNOS, Arg1, and β-actin in BMDM-M2 cells pretreated for 2 h with 1 μM CsA or not before addition of CQ or PBS ($n = 3$). **m** The expression of p-p38, total p38, and β-actin in BMDM-M2 cells pretreated with 1 μM CsA or not for 2 h before addition of CQ or PBS. **n** Immunofluorescent staining of NF-κBp65 in Mcoln1-SGGFP and Mcoln1-SGRNA Raw264.7 cells with or without CQ treatment. Green, NF-κBp65; blue, DAPI. Scale bar, 20 μm. **o** The distribution of NF-κBp65 in the same experimental conditions as **l**. Green, NF-κBp65; blue, DAPI. Scale bar, 20 μm. Data shown are representative of three independent experiments and error bars represent mean ± SEM. *$P < 0.05$; **$P < 0.01$; ***$P < 0.001$ (Student's *t*-test)

antitumor T-cell immunity[46,47]. Notably, TAMs can interact with MDSCs and Treg cells to form a vicious immunosuppressive triangle[47,48]. As CQ treatment can reset macrophages and restore T-cell antitumor activity, we hypothesized that CQ treatment is able to terminate the vicious TAM-MDSC-Treg triangle, thus promoting antitumor T-cell immunity. Using flow cytometric analysis, we found that the number of CD11b+Gr-1hi MDSCs and CD4+CD25+Foxp3+ Treg cells within the tumor microenvironment was remarkably reduced in CQ-treated tumor-bearing mice, compared with those in the control group (Fig. 7a–d). However, this phenomenon was blocked by macrophage depletion (Fig. 7a–d). Murine MDSCs can be classified as

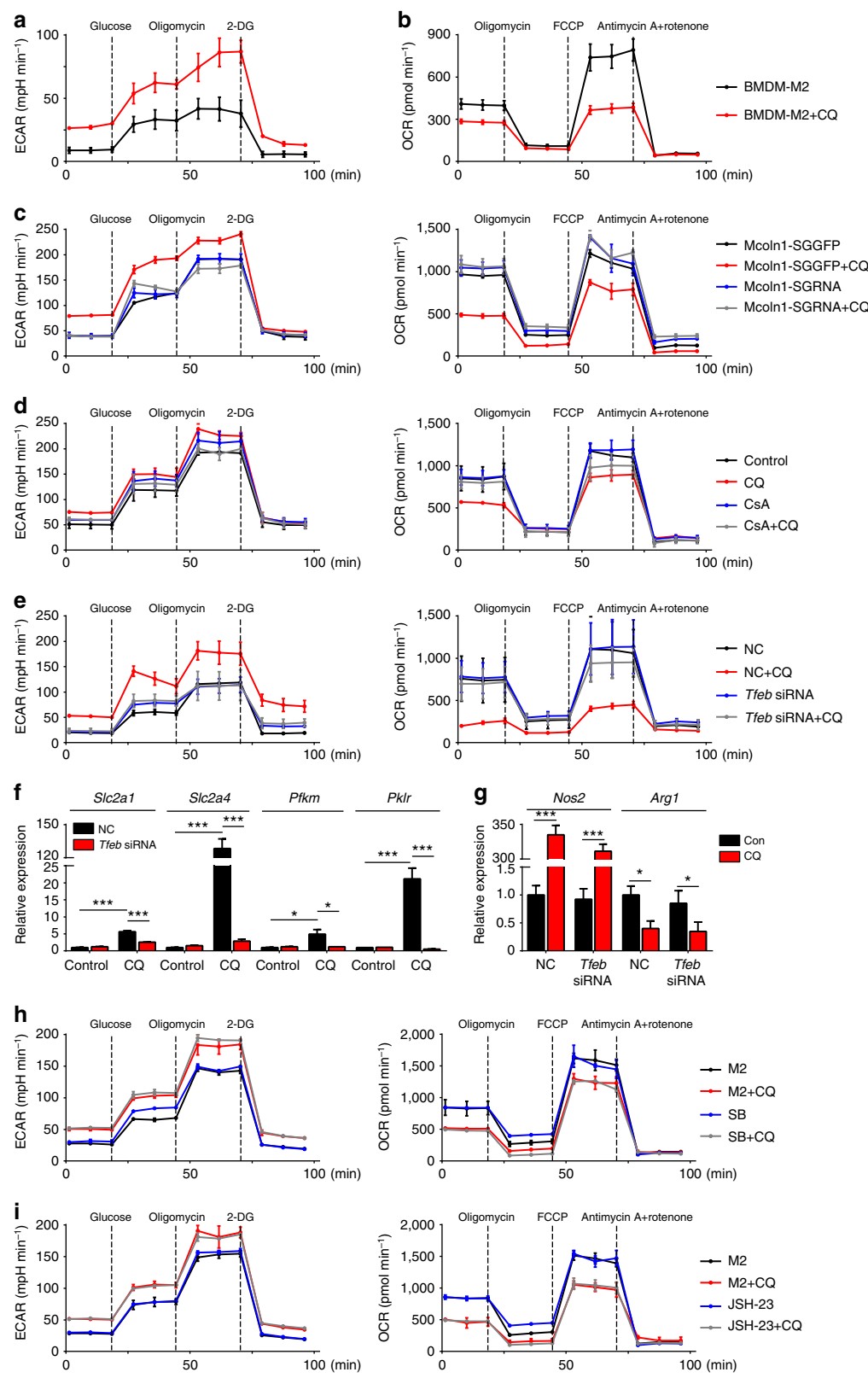

$CD11b^+Ly6C^{hi}Ly6G^-$ monocytic MDSCs and $CD11b^+Ly6C^{lo}$-$Ly6G^+$ granulocytic MDSCs[49–51]. We additionally found that CQ treatment actually decreased both subsets of MDSCs and such CQ-mediated decrease could be disrupted by macrophage depletion (Supplementary Fig. 10a). In addition, injection of CQ-conditioned macrophages into tumor site also resulted in the decrease of MDSCs and Treg cells (Fig. 7e–h and Supplementary Fig. 10b, c). However, such effect was blocked in CQ-conditioned macrophages with Mcoln1 knockout (Fig. 7i and Supplementary Fig. 10d, e), suggesting that CQ-reset macrophages ameliorate tumor immune microenvironment via the lysosomal calcium release-dependent pathway. Moreover, TFEB knockdown also hindered the amelioration of tumor microenvironment by CQ-conditioned macrophages (Fig. 7j and Supplementary Fig. 10f, g). An important means, through which MDSCs exert their immunosuppressive function, lies in their induction of Treg cells in tumor-bearing hosts[52–54]. We subsequently tested CQ combined with low-dose chemotherapeutic drug cyclophosphamide (CY), a well-established method that selectively depletes Tregs[55,56], to treat tumor-bearing mice in order to maximize antitumor immune responses. Interestingly, the combination therapy generated even greater inhibition on tumor growth in B16 melanoma and H22 hepatocarcinoma mouse models (Fig. 7k,l). Consistently, depleting Treg cells by anti-CD25 antibodies also resulted in an enhanced inhibitory effect of CQ on tumor growth (Supplementary Fig. 10h, i). Such inhibitory effect was also observed by Gr-1 antibody-mediated MDSCs depletion (Supplementary Fig. 10j). Together, these data suggest that CQ-reset macrophages ameliorate the immunosuppressive tumor immune microenvironment through a lysosomal calcium-TFEB pathway, and combining other immune therapeutics with CQ can further potentiate its beneficial effect on the immunosuppressive tumor microenvironment. A schematic diagram of CQ resetting macrophages is shown in Fig. 8.

## Discussion

Chloroquine, an old anti-malarial drug, is being exploited in clinical trials for potential anticancer therapy[57]. Its anticancer mechanisms have been assumed to be mediated by curtailing tumor cell autophagy and promoting tumor vessel normalization via targeting tumor endothelial cells[15,16,58]. In the present study, we demonstrate that immune elements actually dominate CQ's antitumor efficacy. We provide clear evidence that CQ exerts its antitumor effect through its immune-regulating function.

Consistent with previous reports[16,59], we confirmed the antitumor efficacy of CQ using B16 melanoma and H22 hepatocarcinoma mouse tumor models. However, such antitumor effect disappears in T-cell-deficient mice, suggesting that CQ mobilizes T-cell immunity for its antitumor effect. Furthermore, CQ's antitumor effect also disappears in macrophage-deficient mice, suggesting that CQ uses macrophages as a means to activate antitumor T-cell immune response. In support of this notion, we found that macrophages in tumor microenvironment are reset by CQ from tumor-promoting M2 to tumor-inhibiting M1 phenotype. M1 macrophages are capable of killing tumor cells through different ways; however, tumor cells can efficiently educate macrophages and make them develop toward M2 phenotype using direct or indirect means. In the tumor microenvironment, various factors such as IL-13, PGE2, macrophage colony-stimulating factor (M-CSF), vascular endothelial growth factor, and lactic acid promote M2 macrophages development[8,60,61]. Moreover, tumor cell-derived microparticles by virtue of their intrinsic biological information can act as a general mechanism to educate macrophages into M2-type TAMs, leading to tumor growth, metastasis, as well as cancer stem cells development[21]. Targeting M2 TAMs in tumor microenvironment is considered as an ideal therapeutic strategy against cancers. Currently, many studies focus on how to deplete M2 TAMs, to block CCL2/CCR2 recruitment pathway or to interfere with M2-related signaling pathways such as $CD47/SIRP\alpha$[62,63]. However, our present findings provide an alternative strategy of targeting M2 TAMs, which possesses several advantages as follows: (1) CQ being a small chemical compound is easily distributed in tumor microenvironment to target TAMs; (2) CQ does not deplete M2 macrophages but instead reset them into M1 phenotype, thus better utilizing macrophages to reconstruct antitumor immune microenvironment; (3) CQ is a very safe drug with a long clinical track record; and (4) clinical use of CQ is very affordable. All in all, our findings provide new insight into the anticancer mechanisms of CQ and identify CQ as a tumor immunotherapeutic agent.

An important finding in this study is that macrophage polarization is regulated by the lysosomes. Phagocytosis of extracellular materials and effectively degrading them in lysosomes is a core task of macrophages. As a membrane-bound organelle, lysosomes are spherical vesicles, which contain various hydrolytic enzymes that can break down all kinds of biomolecules[12]. However, this degrading process is dependent on the acidic pH value inside the lysosomal lumen. Lysosomes maintain pH value ranging from 4.5 to 5.0 by pumping $H^+$ ions from the cytosol into lysosomal lumen via $H^+$-ATPases[11]. M2 macrophages have prominent phagocytotic function while M1 macrophages are mainly linked to inflammation and have decreased phagocytotic ability. Thus, it is reasonable to speculate that CQ promotes M1 polarization of macrophages through interfering with phagocytosis by increasing lysosomal pH. Although lysosomes were for a long time considered as the waste disposal system of the cell, mounting evidence indicates that lysosomes have a much broader function involving cell secretion, plasma membrane repair, signaling, and energy metabolism[12,36]. Notably, lysosomal pH regulates both waste disposal and non-degrading functions of lysosomes. Thus, CQ treatment may alter non-degrading functions of lysosomes by changing lysosomal pH, thereby resetting the phenotype and function of macrophages. One of the consequences of lysosomal pH alteration is calcium release from lysosomes[64]. Recent studies

---

**Fig. 6** $Ca^{2+}$-activated TFEB reprograms macrophage metabolic mode. **a** Extracellular acidification rate (ECAR) of BMDM-M2 cells was measured with or without 10 μM CQ treatment ($n = 3$). **b** Oxygen consumption rate (OCR) of BMDM-M2 cells were measured with or without 10 μM CQ treatment ($n = 3$). **c** ECAR (left) and OCR (right) in Mcoln1-SGGFP and Mcoln1-SGRNA Raw264.7 cells were measured with or without CQ treatment ($n = 3$). **d** ECAR (left) and OCR (right) were measured in Raw264.7 cells pretreated for 2 h with 1 μM CsA or not before addition of CQ or PBS ($n = 3$). **e** ECAR (left) and OCR (right) in Tfeb and control (NC) siRNA Raw264.7 cells were measured with or without CQ treatment ($n = 3$). **f** The mRNA expression of Scl2a1, Slc2a4, Pfkm, and Pklr were analyzed by real-time qPCR in Tfeb and control (NC) siRNA Raw264.7 cells with or without CQ treatment ($n = 3$). **g** The mRNA expression of Nos2 and Arg1 were analyzed by real-time qPCR in Tfeb and control (NC) siRNA Raw264.7 cells that had been treated with PBS or CQ ($n = 3$). **h** ECAR (left) and OCR (left) were measured in BMDM-M2 cells that had been pretreated for 2 h with 10 μM SB203580 or not before addition of CQ or PBS ($n = 3$). **i** ECAR (left) and OCR (left) were measured in BMDM-M2 cells that had been pretreated for 2 h with 10 μM JSH-23 or not before addition of CQ or PBS ($n = 3$). Data shown are representative of three independent experiments and error bars represent mean ± SEM. *$P < 0.05$; **$P < 0.01$; ***$P < 0.001$ (Student's $t$-test)

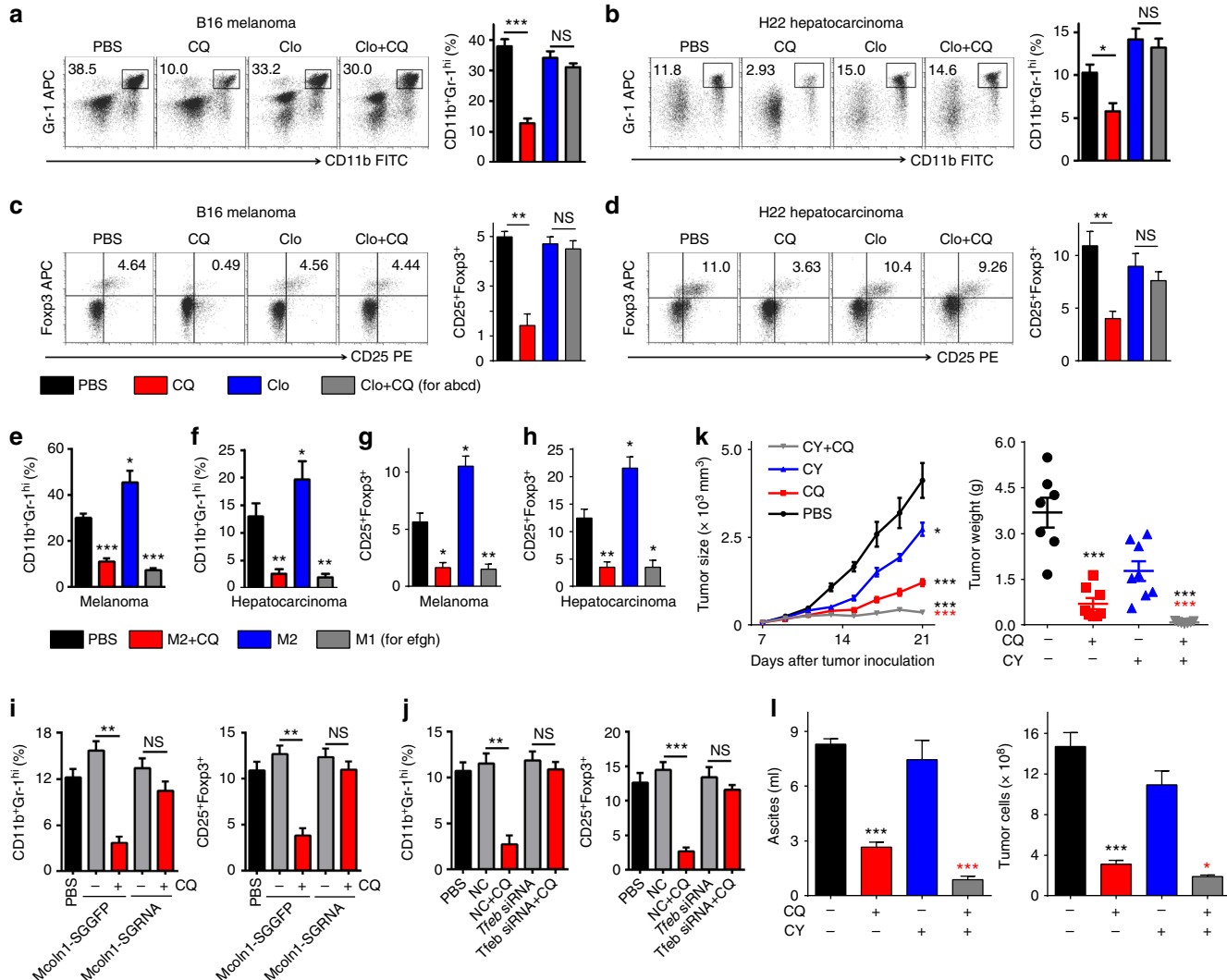

**Fig. 7** CQ-reset macrophages ameliorate tumor immunosuppression. **a, b, c, d** C57BL/6 mice bearing subcutaneous B16 melanoma and BALB/c mice bearing H22 hepatocarcinoma ascites that received clodronate liposomes or control treatment were given i.p. injections with PBS or CQ. Representative flow cytometric analysis and quantification of MDSCs number in melanoma **a** ($n = 6$) and hepatocarcinoma **b** ($n = 6$). Representative flow cytometric analysis and quantification of Treg number in melanoma **c** ($n = 6$) and hepatocarcinoma **d** ($n = 6$). **e, f, g, h** C57BL/6 mice bearing subcutaneous B16 melanoma and BALB/c mice bearing H22 hepatocarcinoma ascites received intratumor injection with either PBS, CQ-treated BMDM-M2, BMDM-M2, or BMDM-M1 cells. MDSCs were quantified in melanoma **e** ($n = 6$) and hepatocarcinoma **f** ($n = 6$); Tregs were quantified in melanoma **g** ($n = 6$) and hepatocarcinoma **h** ($n = 6$). **i** BALB/c mice bearing H22 hepatocarcinoma ascites received i.p. injection with PBS or cell therapy with Mcoln1-SGGFP or Mcoln1-SGRNA Raw264.7 macrophages that had been pretreated with CQ or PBS. Quantification of MDSCs (left) and Tregs (right) were done ($n = 3$). **j** BALB/c mice bearing H22 hepatocarcinoma ascites received i.p. injection with PBS or cell therapy with *Tfeb* or control (NC) siRNA Raw264.7 macrophages that had been pretreated with CQ or PBS. Quantification of MDSCs (left) and Tregs (right) were analyzed ($n = 3$). **k** C57BL/6 mice ($n = 9$) with subcutaneous B16 melanoma were treated with PBS or CQ by intratumor injection along with i.p. administration of 0.5 mg CY or not. The tumor growth was followed (left). After 2 weeks, the mice were killed and the weight of the tumors was recorded (right). Black asterisk, PBS group compared with CQ, CY, and CY + CQ group; red asterisk, CQ group compared with CY + CQ group. **l** BALB/c mice ($n = 6$) bearing H22 hepatocarcinoma ascites were i.p. treated with PBS or CQ together with i.p. administration of 0.5 mg CY or not. Tumor ascites volume (left) and tumor cell number (right) were measured after one week treatment. Black asterisk, CQ group compared with PBS group; red asterisk, CQ group compared with CY + CQ group. Data shown are representative of three independent experiments and error bars represent mean ± SEM. *$P < 0.05$; **$P < 0.01$; ***$P < 0.001$; NS, not statistically significant (Student's *t*-test)

have revealed that in addition to the endoplasmic reticulum, the lysosomal organelles are also important reservoirs for calcium storage[65]. The present study and previous reports together demonstrate that lysosomal pH alteration results in the release of calcium from lysosomes, even though the underlying mechanism has not been elucidated[64]. Importantly, blockading lysosomal calcium release disrupts CQ-induced M1 macrophages polarization, suggesting that lysosomal calcium release is necessary for resetting macrophages from M2 to M1 phenotypes. How does the

released calcium mediate the resetting process? The development of M1 macrophages not only includes the M1 phenotype formation but also involves an intrinsic metabolic change. In this study, we find that the released lysosomal calcium not only activates p38 and NF-κB that mediate M1 phenotype formation, but also activates TFEB, a cytosolic transcription factor that is a critical regulator of lysosome biogenesis and the M1-associated glycolysis. Moreover, the $Ca^{2+}$ signaling-mediated p38, and NF-κB and $Ca^{2+}$ signaling-mediated TFEB are two independent

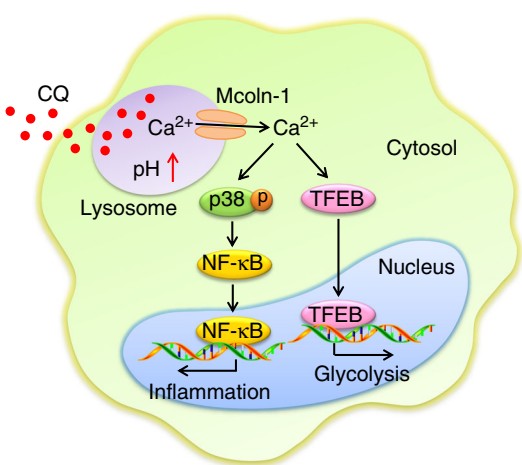

**Fig. 8** The schematic diagram of CQ resetting TAMs. CQ increases TAMs lysosomal pH, causing $Ca^{2+}$ release via the lysosomal $Ca^{2+}$ channel mucolipin 1 (Mcoln1), which induces the activation of p38 and NF-κB, thus polarizing TAMs to produce inflammatory cytokines. Meanwhile, released $Ca^{2+}$ activates transcription factor EB (TFEB) that reprograms macrophage metabolism towards glycolysis

pathways, and both of them are required to form a full M1 macrophage. Together, our findings reveal a surprising role for lysosomes in macrophage polarization.

CQ has exhibited an ability to inhibit or kill tumor cells, which is used to explain CQ's antitumor effect. However, CQ is unable to kill the large bulk of tumor cells and the remnant tumor cells may quickly regrow, thus annulling the direct tumor-killing effect, implying that CQ probably mobilizes other effector function(s) to exert its antitumor effect. In this study, we reveal that CQ potentiates immune reactions against tumors, which is not contradictory to the killing effect of CQ on tumor cells, as the killed tumor cells may release tumor antigens that facilitate antitumor immunity. In this study, we find that CQ promotes antitumor T-cell immunity in a macrophage-dependent manner. However, the key question is how CQ-reset macrophages remodel tumor immune microenvironment, leading to enhanced antitumor immunity. It is well established that immunoregulatory cells, typically including Treg cells and MDSCs, profoundly suppress tumor-specific T-cell immune response. After CQ treatment, we find that Treg cells and MDSCs are significantly reduced in the tumor microenvironment. This undoubtedly relieves the suppressive effect on T cells and facilitates antitumor T-cell immunity. Previously, Maes et al.[16] showed that CQ increases endothelial Notch 1 signaling, leading to tumor vessel normalization. Intriguingly, normalization of aberrant tumor vasculature favors the recruitment of tumor-specific T cells. In this study, we demonstrate that CQ-triggered lysosomal $Ca^{2+}$ signaling is critical to activate two pathways that mediate the polarization of M2 to M1. Lysosomal $Ca^{2+}$ signaling activates p38 and NF-κB for M1 phenotype on one hand and also activates TFEB for M1 metabolism on the other hand. TFEB blockade only influences metabolic switch of macrophages but not their phenotype. However, TFEB-regulated glucose metabolism is required for the function exertion of CQ-reset macrophages. Thus, although TFEB has no effect on the CQ-reset macrophage phenotype, TFEB knockdown seems to be sufficient to thwart CQ-decreased infiltration of MDSCs and Treg cells in tumor microenvironment via the macrophage-dependent pathway. Taken together, we propose that three pathways are used by CQ for exerting its antitumor effect: (1) CQ resets macrophages from tumor-associated M2 into M1 phenotype by mobilizing lysosomal

$Ca^{2+}$ reservoirs and inducing of $Ca^{2+}$ signaling, thus transforming the immunosuppressive tumor microenvironment via reducing the burden of Treg cells and MDSCs; (2) CQ normalizes tumor vessels via regulating endothelial cells, facilitating tumor-specific T-cell infiltration into tumor site where CQ has already relieved the T-cell immunosuppression, leading to efficient tumor cells killing by infiltrated T cells; and (3) CQ partially kills tumor cells directly via disrupting autophagy of tumor cells, leading to the release of tumor antigens and danger signals that further promote antitumor immunity.

In summary, the data in this study clearly show that CQ, by virtue of its ability to increase lysosomal pH value and lysosomal number, resets TAMs from M2 to M1 phenotype, unveiling that CQ actually functions as an immuno-modulator for its antitumor effect. These findings may open a new avenue for cancer immunotherapy.

## Methods

**Cell lines and reagents**. Murine melanoma cell line B16 and OVA-B16, murine hepatocarcinoma cell line H22, and murine macrophage cell line Raw264.7 were purchased from China Infrastructure of Cell Line Resources (Beijing, China) and cultured with RMPI-1640 (Life Technologies) or high glucose-Dulbecco's modified Eagle's medium (Corning) medium supplemented with 10% feral bovine serum (FBS) (Gibco, Australia) at 37 °C with 5% $CO_2$. Cells were tested for mycoplasma detection, interspecies cross-contamination, and authenticated by isoenzyme and short tandem repeat analyses in Cell Resource Centre of Peking Union Medical College before the beginning of the study and spontaneously during the research. Chloroquine, OVA257-264, OVA323-339, and CY were purchased from Sigma-Aldrich. SB203580, JSH-23, Baf, IPI-549, 1400W 2HCl, and CsA were purchased from Selleck. Anti-CD3 (145-2C11), anti-CD8 (53-6.7), anti-CD25 (PC-61.5.3) mouse neutralizing antibody, and isotype control (N/A) were purchased form Bio X Cell. Purified anti-mouse Ly-6G/Ly-6C (Gr-1) antibody (RB6-8C5) and rat IgG2bκ isotype control were purchased from BioLegend. The macrophage depletion reagent, clodronate liposomes, was purchased from FormuMax Scientific USA. Ryanodine and CGP37157 were purchased from Abcam.

**Mice and animal experiments**. Female C57BL/6 (CD45.2), BALB/c, and BALB/c-nude mice (6–8-week-old) were purchased from Center for Experimental Animal Research for studies approved by the Institutional Animal Care and Use Committee of Peking Union Medical College (Beijing, China). OT-I and OT-II TCR transgenic mice were gifted by Dr Hui Zhang (Sun Yat-Sen University, Guangzhou, China). Female CD45.1 C57BL/6 mice were purchased from Laboratory Animal Center of Peking University. In melanoma subcutaneous model, $2 \times 10^5$ B16 cells were subcutaneously implanted into the right flank of female C57BL/6 or BALB/c-nude mice. One week later, mice were randomly divided into different groups and treated with $1 \times$ phosphate-buffered saline (PBS), CQ (75 mg kg$^{-1}$ per day), anti-CD3 neutralizing antibody (400 μg per mice, every 4 days, intraperitoneally (i.p.)), anti-CD8 neutralizing antibody (200 μg per mice, every 4 days, i.p.), anti-CD25 neutralizing antibody (200 μg per mice, every 4 days, i.p.), anti-Gr-1 neutralizing antibody (250 μg per mice, every 4 days, i.p.) or isotype control, clodronate liposomes (100 μl per mice, every 4 days, i.p.), IPI-549 (15 mg kg$^{-1}$ per day, oral gavage), CQ-treated BMDM-M2 or BMDM-M2, or BMDM-M1 cells ($1 \times 10^5$ cells, every 2 days, subcutaneously), and CY (0.5 mg per mice, every 2 days, i.p.), respectively, for the indicated time. In OVA-B16 melanoma subcutaneous model, $5 \times 10^5$ OVA-B16 cells were subcutaneously injected into CD45.1 C57BL/6 mice. Mice were randomly divided into different groups when the tumor volume was about 5 mm × 5 mm and treated with PBS, CQ, and/or adoptive transferred CD8$^+$ OT-I T cells. For H22 subcutaneous hepatocarcinoma in BALB/c-nude mice, $2 \times 10^5$ H22 cells were subcutaneously implanted into the right flank of female BALB/c-nude mice. One week later, mice were randomly divided into two groups and treated with PBS or CQ. Subcutaneous tumor growth was recorded with the length ($L$) and width ($W$) of tumors by vernier calipers, and the tumor size ($V$) was calculated by the formula $V = (L \times W^2)/2$. The growth of subcutaneous model was monitored every 2 days and the survival of tumor-bearing mice was observed every day as well. In melanoma lung metastasis model, $5 \times 10^4$ B16 melanoma cells were intravenously injected into C57BL/6 mice, mice were randomized into two groups after 1 week, and killed to count the lung tumor nodules after 2 weeks i.p. treatment with CQ. In H22 hepatocarcinoma ascites model or hepatocarcinoma muscle model, H22 cells ($5 \times 10^4$) were i.p. implanted into mice or injected into the right thigh muscle of BALB/c mice. On the third day, PBS, CQ, anti-CD3, anti-CD8, anti-CD25, anti-Gr-1-neutralizing antibody or isotype control, clodronate liposomes, IPI-549, CQ-treated BMDM-M2 or BMDM-M2, or BMDM-M1 cells ($1 \times 10^5$ cells, every 2 days), Mcoln1-SGGFP or Mcoln1-SGRNA (SG1, SG2, and SG3) Raw264.7 cells, NC or *Tfeb* small interfering RNA (siRNA) (si#1, si#2, si#3) Raw264.7 cells ($1 \times 10^5$ cells, every 2 days), CY were administrated into the tumor-bearing mice for 1 week, respectively. On day 11, the mice were killed to analyze

the microenvironment of tumor ascites and the muscle tumor weight was measured by the weight of right thigh minus the left.

**Flow cytometry.** For surface marker analysis, live cells were re-suspended in 1× PBS and stained with anti-mouse CD3 (17A2, 100204), CD4 (GK1.5, 100406), CD8a (53-6.7, 100708), CD25 (3C7, 101904), PD1 (RMP1-30, 109112), CTLA-4 (UC10⁻4B9, 106310), CD45 (30-F11, 103106), CD45.2 (104, 109814), CD69 (H1.2F3, 104514), CD107a (1D4B, 121614), CD137 (17B5, 106110), CD80 (16-10A1, 104708), CD86 (GL-1, 105008), MHC-II (M5/114.15.2, 107608), CCR7 (4B12, 120108), CD206 (C068C2, 141708), CD301 (LOM-14, 145704), F4/80 (BM8, 123116), CD11b (M1/70, 101206), Gr-1 (RB6-8C5, 108412), Ly6C (HK1.4, 128008), Ly6G (1A8, 127614), and CD11c (N418, 117308), and anti-human CD163 (GHI/61, 333606), CD206 (15-2, 321110), and CD64 (10.1, 305008) at 4 °C 30 min. The concentration at each antibody was used as the product protocol recommended. All surface flow cytometry antibodies were purchased from Biolegend, and in some cases anti-mouse CD16/32 (93, BioLegend, 101320) were pre-added to block nonspecific binding of immunoglobulin to macrophage Fc receptors. For intracellular cytokine staining, cells were fixed and permeabilized after the stimulation with Cell Stimulation Cocktail (plus protein transport inhibitors) (eBioscience) at incubator for 6 h and labeled with anti-mouse IFN-γ (XMG1.2, Biolegend, 505808), IL-12p40 (C17.8, eBioscience, 12-7123-82), TNF-α (MP6-XT22, Biolegend, 506306), T-bet (4B10, Biolegend, 644810), anti-human IFN-γ (4 S.B3, eBioscience, 12-7319-42), IL-12p40 (C11.5, Biolegend, 501804), TNF-α (MAb11, Biolegend, 502909), TGF-β1 (TW4-2F8, Biolegend, 349604), and IL-10 (JES3-9D7, eBioscience, 25-7108-42) after the surface markers staining. For Foxp3 (FJK-16s, eBioscience, 17-5773-82), iNOS (CXNFT, eBioscience, 12-5920-82), and PE-conjugated anti-Arg1 (R&D Systems, IC5868P) staining, the cells were fixed and permeabilized without stimulation. Data were acquired by BD Accuri C6 or Life Technologies Attune NxT and analyzed with FlowJo software.

**T-cell proliferation and tumor cytotoxicity assay.** T cells were isolated from female C57BL/6 mice spleen by Pan T Cell Isolation Kit (Miltenyi Biotec). After the CFSE (carboxyfluorescein succinimidyl ester) (Sigma-Aldrich) labeled, T cells were cultured with 10 ng ml⁻¹ IL-2 (PeproTech) in completed RPMI-1640 medium (10% FBS, 100 U ml⁻¹ penicillin–streptomycin) and stimulated with Dynabeads Mouse T-Activator CD3/CD28 beads (Life Technologies). For specific T-cell proliferation, splenocytes from OT-I or OT-II TCR transgenic mice were labeled with CFSE and then stimulated with OVA257-264 peptide or OVA323-339 peptide, respectively, in the presence or absence of conditioned medium from control or CQ-treated BMDM-M2 cells. After 72 h, CFSE was detected by BD Accuri C6 and T cells were labeled with CD8 or CD4 in the specific T-cell proliferation cases. For T-cell-based OVA-B16 apoptosis assay ex vivo, adoptive transferred CD45.2 OT-I CD8⁺ T cells were sorted by CD45.2⁺ cells with BD ARIA III from PBS or CQ-treated OVA-B16-bearing CD45.1 mice tumor tissues and co-cultured with OVA-B16 cells at a ratio of 30 : 1 for cytotoxicity assay. The tumor cell apoptosis were measured by CD3⁻Annexin-V⁺ after 24 h. In some cases, macrophage-induced tumor cell apoptosis or CQ- and Baf-induced macrophage apoptosis were measured by Annexin-V and PI staining (BD Pharmingen).

**Generation of mouse bone marrow-derived DCs.** BMDCs were acquired as described before[66]. In brief, bone marrow cells were collected from female C57BL/6 mice femurs and cultured in six-well plates with 20 ng ml⁻¹ granulocyte M-CSF (PeproTech), and 20 ng ml⁻¹ IL-4 (PeproTech). On day 3 and 5, the medium were half-renewed and then the non-adherent cells were collected for experiments.

**Generation of mouse BMDM and human macrophages.** As we described before[21], the bone marrow cells were isolated from female C57BL/6 mice femurs and cultured with 20 ng ml⁻¹ recombinant M-CSF (PeproTech) for 5 days. On day 6, naive macrophages (BMDMs) were collected and then stimulated with 20 ng ml⁻¹ IL-4 (PeproTech) or 100 ng ml⁻¹ LPS (Sigma-Aldrich) plus 20 ng ml⁻¹ IFN-γ (PeproTech) to generate the BMDM-M2 or BMDM-M1 macrophages, respectively, for 24 h. For human macrophage culture, monocytes were isolated from healthy donors' blood by Human Monocyte Enrichment Cocktail (STEMCELL Technologies) and cultured with 20 ng ml⁻¹ recombinant human M-CSF (PeproTech) to induce human macrophages. Seven days later, 20 ng ml⁻¹ recombinant human IL-4 (PeproTech) was added for human macrophages polarization. The study was approved by the Clinical Trial Ethics Committee of Peking Union Medical College. All healthy donors have provided written informed consent to participate in the study.

**Tumor-infiltrating lymphocyte and macrophage isolation.** Tumor tissues were collected and cut into small pieces in PBS. After centrifugation, enzyme digestion was performed with 1 mg ml⁻¹ collagenase (Sigma-Aldrich), 2 units ml⁻¹ hyaluronidase (Sigma-Aldrich), and 0.1 mg ml⁻¹ DNase (Sigma-Aldrich) for 1 h. Then the cell suspension was centrifuged with Ficoll to get the mononuclear cells and/or sorted with anti-F4/80 microbeads (Miltenyi Biotec) to get tumor-infiltrating macrophages. For tumor-infiltrating lymphocyte and macrophage isolation in tumor ascites, ascites volume and the CD45⁻ cells (H22 tumor cell number) were counted, ascites were centrifuged with Ficoll directly to get mononuclear cells, and then sorted with anti-F4/80 microbeads (Miltenyi Biotec) to get tumor-infiltrating macrophages.

**Real-time qPCR.** Total RNA (1 μg) were extracted from cells or tumor tissues with TRIzol reagent (Invitrogen, USA) and reverse-transcribed into cDNA by ReverTra Ace Kit (Toyobo). The cDNA were amplified by the THUNDERBIRD SYBR qPCR Mix (Toyobo) on a StepOnePlus Real-Time PCR System (Thermo Fisher Scientific). The mRNA levels were normalized by β-actin. The primer sequences are shown at Supplementary Table.

**Chromatin immunoprecipitation–qPCR.** ChIP–qPCR was conducted as the procedure of MAGnityTM Chromatin Immunoprecipitation System (Invitrogen). Briefly, BMDM-M2 and CQ-treated BMDM-M2 cells were crosslinked and chromatin was extracted and sheared. Immunoprecipitation was performed by using anti-TFEB (Bethyl Laboratories, A303-673A). The primer sequences used for ChIP–qPCR are shown as follows: Slc2a1, 5′-GCAAAGTGGTGATCAGGAGA-GAG-3′ (sense) and 5′-TCCCATCTAGGTTCACCTGAAGG-3′ (antisense); Slc2a4, 5′-TGGCCAATGGGTGTTGTGAAGG-3′ (sense) and 5′-AAA-GACTCAGGCGCTGCAAT-3′ (antisense); Pfkm, 5′-CCTGGGACACA-GAGTAAAGTTGA-3′ (sense) and 5′-AGCAGGTTGCAAGAGTCTGGG A-3′ (antisense); Pklr, 5′-GAAGGATGCCCACTACAGCCTC-3′ (sense) and 5′-TCT GCCTTTGTCAGTGGGATGG-3′ (antisense).

**ELISA and NO level assay.** Enzyme-linked immunosorbent assay (ELISA) was performed as the manufacturer's procedure. Human and mouse IFN-γ, TNF-α, and IL-10 ELISA kits were purchased from PeproTech. Human and mouse TGF-β1 and IL-12p40/p70 ELISA kits were purchased from R&D Systems. For NO level assay, total nitrite/nitrate was measured in cell lysate by using Nitric Oxide Colorimetric Assay Kit (BioVision, K262) according to the suppler's instructions.

**Western blot analysis.** All cell lysates and Prestained Protein Ladder (Thermo Fisher Scientific) were separated by 10% SDS-polyacrylamide gel electrophoresis and then transferred into nitrocellulose membranes. After blocking with 5% bovine serum albumin (BSA) in Tris-buffered saline containing 0.1% Tween-20 for 1 h at room temperature, the membranes were incubated with specific anti-Arg1 (9819, 1 : 1,000), anti-iNOS (2982, 1 : 1,000), anti-p-p38 MAPK (4511, 1 : 1,000), anti-p38 MAPK (8690, 1 : 1,000), anti-p-JNK (4668, 1 : 1,000), anti-JNK (9252, 1 : 1,000), anti-p-Erk1/2 (4370, 1 : 1,000), anti-Erk1/2 (4695, 1 : 1,000), anti-NF-κBp65 (8242, 1 : 1,000), anti-β-actin (3700, 1 : 1,000), and anti-Histone H3 (4499, 1 : 2,000) from Cell Signaling Technology, and anti-TFEB (Bethyl Laboratories, A303-673A, 1 : 500) overnight at 4 °C. Then the membranes were washed and incubated with horseradish peroxidase-conjugated secondary antibodies. The Nuclear/Cytosol Fractionation Kit (BioVision) was used according to the manufacturer's procedure, in some cases, to extract the cytoplasmic and nuclear proteins. Then the proteins were visualized by ECL western blotting reagent (Thermo Pierce). Uncropped images of western blottings are provided in Supplementary Fig. 11, 12, and 13.

**Immunofluorescence and immunohistochemical staining.** For immuno-fluoresence staining, cells were seeded in confocal dish for 24 h with or without CQ treatment and CsA pretreatment in some cases. The cells were fixed with 4% paraformaldehyde in PBS pH 7.4 for 10 min at room temperature and then permeabilized with 0.5% Triton X-100 in PBS for 10 min. After blocking with 2% BSA in PBS containing 0.1% Tween-20 for 30 min, cells were incubated with NF-κBp65 (Cell Signaling Technology, #8242, 1 : 400), TFEB (Bethyl Laboratories, #A303-673A, 1 : 500), Lamp1 (Abcam, ab25245, 1 : 500), and Lamp2 (Abcam, ab25339, 1 : 100) in 2% BSA in PBS containing 0.1% Tween-20 overnight at 4 °C. After washing and staining with secondary antibody for 1 h at room temperature, nucleus were stained with DAPI (4′,6-diamidino-2-phenylindole) (2 μg ml⁻¹). The merged figures were analyzed by OLYMPUS two-photon microscope. For immunohistochemical staining, melanoma tissues were isolated from tumor-bearing mice and fixed in 37% formalin and embedded in paraffin. Next, the sections was incubated with the anti-Granzyme B (Abcam, ab4059, 1 : 1,000) by using the UltraVision Quanto Detection System kit (Thermo Scientific) and observed by using the DAB Quanto kit (Thermo Scientific).

**Lysosomal pH value assay.** For lysosomal pH value measurement of BMDM-M1, BMDM-M2, and CQ-treated BMDM-M2 cells, cells were trypsinized and collected (1 × 10⁶ cells per ml) and then stained with LysoSensor Yellow/Blue DND-160 (1 : 1,000 diluted and pre-incubated in complete culture medium for 30 min at 37 °C; Thermo Fisher Scientific, L7545) for 3 min at 37 °C. After washing with ice PBS, cells were transferred into a black 96-well plate (1 × 10⁶ cells per 200 μl per well). The fluorescence intensity was measured on the Synergy H1 (BioTek) at Ex-360/Em-440 and Ex-360/Em-550 with 10 μM of valinomycin and 10 μM of nigericin added. The standard curve of lysosomal pH value was measured by Intracellular pH Calibration Buffer Kit (Thermo Fisher Scientific, P35379). For LysoSensor Green probes (Thermo Fisher Scientific, L7535) staining in BMDM-M1 and M2 cells, cells were incubated with probes for 30 min to 1 hour, then collected and washed with PBS to measure the mean fluorescence intensity by flow cytometry.

**Intracellular calcium concentration assay**. Cells were incubated with 5 μM Fluo-4 AM (Thermo Fisher Scientific, F14201) in PBS. Calcium concentration was determined by the Fluo-4 AM mean fluorescence intensity with ionomycin stimulated (Cayman, 10004974) on a real-time live-cell imaging with a high-speed laser confocal microscope.

**Transmission electron microscope**. BMDM-M2 and CQ-treated BMDM-M2 cells were collected and washed with PBS for three times, then fixed with 2.5% glutaraldehyde at 4 ℃ overnight. After fixation, cells were washed twice with 0.1 M cacodylate buffer and kept in the cacodylate buffer overnight at 4 ℃. The cells were rinsed with 0.1 M cacodylate buffer for 10 min twice, fixed with 1% OsO₄ in cacodylate buffer for 100 min at room temperature, and rinsed with water for three times. The cells were then dehydrated by a graded ethanol solution process, and infiltrated with a series of embedding resins. Then the cells were embedded with Spur's resin at 60 ℃ overnight. The ultrathin sections were analyzed with a JEM1010 electron microscope (JEOL, Japan).

**The ECAR and OCR analysis**. Seahorse XF24 Extracellular Flux Analyzers (Agilent Technologies) was used to measure the ECAR and OCR. Briefly, with or without CQ-treated cells were seeded in XF24 plates, then the plates were detected according to the manufacturer of XF Glycolysis Stress Test Kit (103020-100) and XF Cell Mito Stress Test Kit (103015-100).

**CRISPR/Cas9 plasmid construction and transfection**. The CRISPR/Cas9 based on pX330 plasmid was constructed. The guide RNA sequence are as follows: SGGFP: 5′-GGGCGAGGAGCTGTTCACCG-3′; p38-SGRNA1: 5′-TGG TCTTGTTCAGCTCCTGC-3′; p38-SGRNA2: 5′-GCTGAACAA-GACCATCTGGG-3′; p38-SGRNA3: 5′-ATAGGCGCCCGAGCCCACCG-3′; Mcoln1-SGRNA1: 5′-AG AAGTACTTGAGGCGGCGG-3′; Mcoln1-SGRNA2: 5′-AGGGCTTGCGGCCTTTGGCC-3′; and Mcoln1-SGRNA3: 5′-GCAT-CAGCTTGCAGGGCTTG-3′. For CRISPR/Cas9 plasmid transfection, Raw264.7 cells were transfected according to the procedure of SF Cell Line 4D-Nucleofector X Kit by the 4D-Nucleofector (Lonza, Switzerland) and sorted by puromycin to acquire monoclonal cell line. For Tfeb and Atp6v1c1 siRNA transfection, it was guide by typical RNAiMAX Transfection Procedure (Life Technologies). The siRNA target sequence of mice Tfeb are as follows: 5′-TGTCTAGCAGCCACCT-GAACGTGTA-3′ (1); 5′-ACAGTCCCATGGCCATGCTACATAT-3′ (2); 5′-GACCTGACTCAGAAGCGAGAGCTAA-3′ (3); Atp6v1c1: 5′-CCAGTGGGA-TATGGCTAAA-3′ (1); and 5′-GCACGAGTTGTACAAGCAT-3′ (2).

**Statistical analysis**. Results were presented as mean ± SEM and statistical significance was examined by an unpaired two-tailed Student's t-test by the Graphpad 6.0 software. Log-rank (Mantel–Cox) test was used to analyze long-term survival curve. The $P$-value < 0.05 was considered as statistically significant.

**Data availability**. The authors declare that all the data supporting the findings of this study are available within the article and its Supplementary Information files, and from the corresponding author on reasonable request.

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

## Acknowledgements

This work was supported by National Natural Science Foundation of China (81472653, 81788101, 81773062, 81661128007, and 81530080), National Basic Research Program of China (2014CB542103), National Natural Science Fund for Young Scholars of China (81502473), CAMS Initiative for Innovative Medicine (2016-I2M-1-007).

## Author contributions

B.H. conceived the project. D.C., J.X., R.F., W.D., X.L., J.L., X.J., J.L., S.M., T.Z., F.C., Y.Z., H.Z., K.T., J.M., and Y.L. performed the experiments. B.H., D.C., J.X., R.F., W.D., X.L., and J.L. developed methodology. B.H., D.C., J.X., X.J., J.L., S.M., T.Z., F.C., Y.Z., H.Z., K.T., J.M., and Y.L. performed data analyses. B.H., D.C., and R.F. wrote the manuscript with input from all authors.

## Additional information

**Competing interests:** The authors declare no competing financial interests.

