## [Peer Review File · Nature Communications]

Reviewer #1 (Remarks to the Author):

The authors used chloroquine (CQ) to repolarize tumor macrophages to an M1 phenotype. This is an interesting approach, although not entirely new (see ref. 3, which should be more clearly acknowledged as a previous paper applying the same concept, and other earlier work). As a mechanism, the authors propose that CQ induces calcium release from lysosomes, which repolarizes macrophage metabolism in several ways: (1) calcium release through mucolipin-1, (2) nuclear translocation of TFEB resulting in (3) increased glycolysis as demonstrated by Seahorse, (4) p38 phosphorylation resulting in (5) nuclear translocation of NFkB, a known M1 polarizer. Thus, there is a convincing pathway from CQ effects causing less acidic lysosomes to the M1 polarization.

1. The authors propose that chloroquine is better than macrophage depletion by clodronate liposomes. This comparison (CQ vs Clo) must be tested statistically, correcting for multiple comparisons. If the difference is not significant, the overall concept is of questionable importance for cancer.
2. Human macrophages do not express much iNOS, so other markers are commonly used for M1. The key in vitro experiments should be repeated with human macrophages, and their polarization assessed. This is important for clinical translation.
3. Figure 1c is underpowered (only 6 mice per group).
4. Figure 2 shows some reduction in PD1 and CTLA4 expression, but both reductions are not complete. Therefore, the chloroquine should be tested in combination with the inhibitor of PI3Kgamma, a known M1 polarizer (ref. 3). In the same vein, only 15-20% of TAMs express IL-12 or IFN γ after CQ treatment. What about the others? Do they at least lose arginase?
5. Figure 4f requires a low-power overview to appreciate how many cells switch on NFkB
6. Figure 5l is lacking untreated and scrambled controls.
7. How does 10 μ M CQ (the in vitro dose) correspond to 75 mg/kg (the clinical dose)?
8. Figure 2f and g: How was the CQ effect assessed here? The Clo treatment should have depleted the cells, so what was actually measured?
9. The less acidic pH in lysosomes of M1 macrophages requires a reference.
10. Line 232, "however" makes no sense here
11. Throughout the manuscript, some issues with grammar, word use and sentence structure.

Reviewer #2 (Remarks to the Author):

The manuscript by Chen et al describes a role for lysosome pH in the regulation of M1 macrophage polarization. The authors show that some of the tumor suppressive effects of treating B16 melanoma or H22 hepatocellular carcinoma cells in vivo with the lysosome inhibitor, chloroquine is mediated by promoting a switch of tumor associated macrophages from the M2 to M1 tumor killing phenotype. CQ mediated M1-switching is shown to correlate with decreased MDSC and Treg infiltration and increased anti-tumor T cell function in vivo. Mechanistically, the authors show that decreasing lysosome pH following CQ treatment leads to MCOLN1 mediated Ca²⁺ release which correlates with increased p38 and NFκB signaling. In addition, the authors provide evidence that Ca²⁺ release is required for TFEB nuclear translocation and metabolic reprogramming of M1 switched macrophages.

This study highlights a potentially important function of the lysosome in non-tumor cell populations. It also highlights the potential dual benefit of treating tumors with chloroquine given the potential for combined suppressive effects on the tumor cells as well as infiltrating macrophage populations. However clarification of some comments outlined below, would provide further clarity and novelty to this study.

-The authors only use CQ in all experiments. It would be valuable to show that another lysosome inhibitor – eg. Bafilomycin A1 – or knockdown of a V-ATPase subunit - also has the same effect.

-TFEB is a known master regulator of lysosome biogenesis and function. Have the authors checked whether TFEB is responsible for the increase in lysosome biogenesis shown in Fig. 5c? Is there an increase in lamp1/2 staining in these cells?

-What are the relative expression levels of lysosome genes (or CLEAR element genes) in CQ treated macrophages?

-Does TFEB KD impact lysosome gene expression, number, or pH?

-KD of TFEB was not able to promote switching from an M2 to M1 state (fig 6g). However KD of TFEB in CQ conditioned macrophages was sufficient to prevent a decrease in infiltrating MDSCs and Treg (fig 7j). How do the authors reconcile these results?

-Do the genes shown to be dependent on TFEB regulation (SLC2A1, SLC2A4, PFKM, PKLR) contain TFEB binding elements in their promoter (CLEAR elements)? Can the authors provide CHIP data to show direct regulation?

-Nuclear localization is not obvious in panels 5e, l, n. Can the authors provide cytoplasmic vs nuclear fractionation via western blot to show a definitive change in cellular localization upon CQ treatment?

-Level of IL-10 (M2) secretion in fig 3g could also be included.

-some spelling and grammatical errors were found in the text.

Reviewer #3 (Remarks to the Author):

The manuscript by Chen et al, describes a novel anti-tumoral action of chloroquine (CQ) that entails the switching of TAMs from M2 to a M1 tumor-killing phenotype supporting anti-tumor immunity responses, by reducing infiltration of immunosuppressive Tregs and MDSCs. The authors analyse the mechanisms of the effect of CQ on TAMs and find that it involves a Ca²⁺-mediated TFEB-dependent pathway reprogramming TAM metabolism. The study is relevant for our understanding of the in vivo effects of CQ. However, several aspects of this study remains to be further validated experimentally in order to support author's conclusions.

General remarks:

In all ex vivo or in vitro experiments with immune cells that the authors have presented, the authors have completely excluded the role of cancer cells. CQ can exert cytotoxic and immunosuppressive effects on or via cancer cells and these effects also play a role in immunological modulation. Considering that a tumor is usually >90% cancer cells, the authors cannot exclude the immunological effects of CQ-treated cancer cells in their in vitro or in vivo experiments. Thus the immune cell co-

culture experiments with DCs or macrophages presented in this manuscript should be repeated in presence of cancer cells as well (B16 and hepatocellular carcinoma) in order to rule in or rule out the contributory or inhibitory roles of cancer cells.

The authors argue that CQ enhances tumor killing by resetting TAMs towards M1 but they don't provide any data on immune-mediated tumor cell killing. For macrophages, simply showing increase in iNOS levels does not mean that the macrophages are killing cancer cells via respiratory burst. This actually needs to be shown (cancer cell death in presence of CQ+macrophages or only macrophages and only CQ, plus, NO levels in media to start with) and iNOS or NO activity should be blocked to establish this. Also T cell-based cancer cell killing analysis should be performed. The authors moreover present no analysis on perforin-granzyme positivity of T cells *in vivo*, or whether T cells derived from CQ-treated tumors can kill B16 cells *ex vivo*/ *in vitro*. All these assays are necessary to validate conclusions of the authors (including the title of this paper).

Specific points:

Figure 1. There is a discrepancy in the data comparing immunocompetent mice with immunodeficient ones. Why are immunocompetent mice always treated with higher dose of CQ (75mg/kg) whereas immunodeficient mice or nude mice are always treated with lower dose (50 mg/kg) of CQ? Similar doses of CQ should be used in both mice cohorts in order to rule out that differences in tumor growth are because of different doses of CQ rather than T cell dependency.

Figure 1. CD3 depletions assays need more statistical analysis to be valid. No statistics are shown for CQ+CD3 condition vs CQ alone. Is the effect of CD3 antibodies even significant? CQ only conditions has no error bars in some places? That is improbable.

Also no interpretation provided for why CD3 depletion has different effect on PBS and CQ conditions?

The assay with splenic T cells that the authors carry out *ex vivo* to rule out direct effect of CQ on T cells is not representative of this effect. Splenic T cells should be derived from the immunocompetent mice treated in Fig 1 with CQ (75mg/kg) and these should be analyzed for markers of cell death (Annexin V-PI), activation (CD69, LAMP1, CD137) and T cell anergy (PD1, CTLA4).

Fig 2. The T cell immunophenotyping is not correctly done. CD8+IFN γ + cells include both CD8+T cells and NK cells - the authors need to quantify CD45+CD3+CD8+IFN γ + population for CD8+T cells. Moreover it is not clear what is the physiological relevance of results in Fig 2d,e. that is - 4% increase in CD8+IFN γ + cells is not really a huge amount for enforcing CD8+T cell-driven immunity. The

authors should better deplete CD8+ cells with antibodies to confirm the relevance of this population in this system. Also it is not clear what population is labelled with CTLA4 or PD1? More markers should be used for T cells, macrophages/myeloid cells and cancer cells to differentiate between different tumoral populations expressing these immune-checkpoints.

The correlation between IFN γ and PD1/CTLA4 levels makes no sense. Sustained IFN γ production up-regulates PD1/CTLA4 as a negative feedback loop mechanism, this is the whole basis of the concept of immune-checkpoints, whereas the authors see the opposite of this. This data needs to be clarified better. More later time-points should be added and direct effects of CQ on PD1/CTLA4 should be better presented.

IL12p40 analysis is not a correct marker for macrophage activation. IL12p70 is the actual active subunit of IL12. While IL12p40 is generally easily up-regulated, in absence of IL12p70, IL12p40 cannot enforce any immunological activity. Analysis for IL12p70 should also be provided and to support author's conclusions, it should follow the same trend as IL12p40.

The authors have analyzed the level of immunostimulatory M1 cytokines but not those of immunosuppressive M2 cytokines. IL10, TGF β and IDO1 levels should also be analyzed in media of M1 and M2 assays to clarify functional M2 polarisations and address the possibilities of partial functional suppression within M1 macrophages.

Fig 3. 10-15% positivity of total macrophages for respective cytokines isn't very physiologically high. The authors need to show actual concentrations of these cytokines in the medium and these concentrations need to (significantly) exceed 10 pg/mL to be physiologically relevant.

Fig 3. Why are macrophages in Fig 3b already more than 60% positive for CD80? This means that the macrophages are pre-stimulated on a certain level. The authors need to optimize the macrophage cultures to have CD80 positivity below 20%.

Fig 7. CD11b+Gr1(hi) are not a specific markers for MDSCs. These markers mark all neutrophils and granulocytes. In order to have a valid MDSC panel the authors need to have CD11b+GR1(hi)CD86(lo)MHCII(lo) wherein low expression of CD86 and MHCII is marker of the possible suppressive nature of these granulocytes. Also MDSCs, are not just granulocytic in origin but also myeloid-MDSCs need to be analyzed by a similar strategy. The authors need to deplete MDSCs in vivo to prove their conclusions regarding MDSCs.

Cyclophosphamide is not a selective Treg depleting methodology, it has several other effects on the tumor (e.g. immunogenic cell death) and gut microbiota (Th17 anti-tumor axes). These data should be replaced by Tregs depleted in the current system via anti-CD25 antibodies. Finally clodronate liposome-based depletion is not specific for macrophages only but targets all phagocytes.

RESPONSES TO REVIEWERS

We would like to express our sincere thanks to all three reviewers for their critical and constructive comments. We have performed substantial additional experiments to address their concerns. We respond point-by-point to each of their comments and criticisms. We feel that their comments have helped us on significantly improving and strengthening the manuscript and clarifying some issues. We hope that the revision has addressed their major concerns.

RESPONSE TO REVIEWER #1:

Reviewer #1 (Remarks to the Author):

The authors used chloroquine (CQ) to repolarize tumor macrophages to an M1 phenotype. This is an interesting approach, although not entirely new (see ref. 3, which should be more clearly acknowledged as a previous paper applying the same concept, and other earlier work). As a mechanism, the authors propose that CQ induces calcium release from lysosomes, which repolarizes macrophage metabolism in several ways: (1) calcium release through mucolipin-1, (2) nuclear translocation of TFEB resulting in (3) increased glycolysis as demonstrated by Seahorse, (4) p38 phosphorylation resulting in (5) nuclear translocation of NFkB, a known M1 polarizer. Thus, there is a convincing pathway from CQ effects causing less acidic lysosomes to the M1 polarization.

1. The authors propose that chloroquine is better than macrophage depletion by clodronate liposomes. This comparison (CQ vs Clo) must be tested statistically, correcting for multiple comparisons. If the difference is not significant, the overall concept is of questionable importance for cancer.

Response:

We thank the reviewer's constructive comment. CQ is an alkaline agent that can increase lysosomal pH value of M2 macrophages. As a result, CQ does not deplete M2 macrophages but instead reset them into M1 phenotype. In this regard, transition of M2 to M1 not only relieves the tumor-promoting effect (M2 depletion) but also acquires new tumor-inhibiting effect (M1 generation), thus theoretically better than single macrophage depletion.

In our original data, we showed that macrophages depletion led to the disruption of the inhibitory effect of CQ on B16 melanoma as well as H22 hepatocarcinoma (Fig. 2a-c). In the revised manuscript, we compared the statistics between the CQ and Clo two groups. We found that the effect of CQ was statistically better than the effect of Clo on B16 melanoma (Fig. 2a) and on H22 hepatocarcinoma ascites (Fig. 2c), suggesting that CQ-based strategy is better than macrophage depletion. In addition, in H22 muscle hepatocarcinoma model, we did not observe the statistical difference between CQ and Clo (Fig. 2b). This might be due to the tissue specificity. The muscle tissue intrinsically produces abundant lactic acid, and the latter effectively polarizes macrophages to an M2-like state (*Nature.2014;513:559-63.*). Thus, even if CQ effectively educates M2 to M1 in

H22 muscle hepatocarcinoma, the endogenous lactic acid may impair this process by re-educating M1 back to M2.

According to the reviewer's suggestion, **we added the statistical asterisk in the revised Figure 2a and 2c.**

2. Human macrophages do not express much iNOS, so other markers are commonly used for M1. The key in vitro experiments should be repeated with human macrophages, and their polarization assessed. This is important for clinical translation.

Response:

In the revised manuscript, we isolated monocytes from healthy donors' blood and cultured them with 20 ng/mL human M-CSF to induce the differentiation toward macrophages. Seven days later, we added 20 ng/mL human IL-4 for M2-like polarization. Then, we used CQ to treat such human M2-like macrophages. These treated cells were analyzed by flow cytometry (CD163, CD206, CD64, IFN- γ , TNF- α , TGF- β 1, IL-10) and real-time qPCR (*Arg1*, *IFN- γ* , *IL-12p40*, *TNF- α* , *TGF- β 1*, *IL-10*), and the supernatants were used for ELISA (IFN- γ , TNF- α , IL-12p40, IL-12p70, TGF- β 1, IL-10). Consistently, CQ treatment also induced human M2 macrophages transiting to M1, as evidenced by upregulation of IFN- γ , TNF- α , IL-12p70, but downregulation of CD163, CD206, CD64, TGF- β 1 and IL-10.

According to the reviewer's suggestion, **we added these results in the revised manuscript, page 9 line 219 and revised Supplementary Figure 4h,i,j.**

3. Figure 1c is underpowered (only 6 mice per group).

Response:

We repeated the B16 lung metastasis experiment with CQ treatment. Eight mice were additional used in each group. Consistent with our original data, CQ markedly decreased the number of tumor nodules in the lungs.

According to the reviewer's suggestion, **we revised Figure 1c (n=14).**

4. Figure 2 shows some reduction in PD1 and CTLA4 expression, but both reductions are not complete. Therefore, the chloroquine should be tested in combination with the inhibitor of PI3Kgamma, a known M1 polarizer (ref. 3). In the same vein, only 15-20% of TAMs express IL-12 or IFN γ after CQ treatment. What about the others? Do they at least lose arginase?

Response:

PI3K γ has been shown to be highly expressed in tumor-associated myeloid cells and inhibition of PI3K γ activity by IPI-549 restores the sensitivity to immune checkpoint blocking therapy in mouse tumor model (*Nature.2016;539:443-7.*). Here, we combined CQ with PI3K γ inhibitor IPI-549 to treat B16 melanoma and H22 hepatocarcinoma. We found that the inhibitory effect of CQ on the growth of B16 melanoma and H22 hepatocarcinoma was enhanced by IPI-549, concomitant with the further decreased PD1 and CTLA-4 expression, suggesting that PI3K γ inhibitor has a synergistic effect on CQ-triggered antitumor immunity. According to the reviewer's suggestion, **we added these results in the revised manuscript, page 9 line 223 and revised Supplementary Figure 4k,l,m.**

In our original Fig. 3a, we used CQ to treat IL-4-conditioned M2 macrophages, and found a

significant upregulation of M1-related cytokine expression (IFN- γ , IL-12p40 and TNF- α). The proportion of these expressing cells is around 15-20%. This might be due to that conversion of M2 to M1 is a slow process and the macrophages were at different stages. In the revised manuscript, we used APC to label IL-12p40, IFN- γ and used PE to label arginase1, and found that CQ treatment resulted in 30% arginase1⁺ converting to arginase1⁻ among IL-12⁻ and IFN- γ ⁻ macrophages.

According to the reviewer's suggestion, **we added these results in the revised manuscript, page 9 line 201 and revised Figure 3f.**

5. Figure 4f requires a low-power overview to appreciate how many cells switch on NFkB.

Response:

In the revised manuscript, we repeated the CQ induced NF- κ B nuclear translocation in M2-like macrophages and presented a low-power view picture, indicating that 90-95% cells switch on NF- κ B.

According to the reviewer's suggestion, **we added these results in the revised manuscript, page 11 line 251 and revised Figure 4f.**

6. Figure 5l is lacking untreated and scrambled controls.

Response:

In the original Figure 5l, we actually used "+" and "-" to show CQ treatment and the untreated control, respectively, and used "Mcoln1-SGGFP" to show the scrambled control. In the revised manuscript, we remarked the Figure legend of Figure 5n to more clearly indicate untreated and scrambled controls.

7. How does 10 μ M CQ (the in vitro dose) correspond to 75 mg/kg (the clinical dose)?

Response:

In our *in vitro* study, we used 10 μ M CQ to treat macrophages, resulting in effectively converting macrophages from M2 to M1 phenotype. However, when we increased the concentration to 20 μ M, we found that some macrophages underwent apoptosis. Thus, we used 10 μ M for *in vitro* experiments.

CQ has been widely studied before. It has been reported that 50mg/kg CQ administration results in 3-13 μ M blood concentration (*Antimicrob Agents Chemother.*2011;55:3899-907; *Parasite.*1994;1:219-26.). Therefore, in our mouse model, we used 75mg/kg CQ to correspond to 10 μ M *in vitro* concentration. In addition, in our study we found that 100mg/kg CQ caused obviously side effect in mice.

According to the reviewer's suggestion, **we described this dosage in the Result section of the revised manuscript, page 5 line 104.**

8. Figure 2f and g: How was the CQ effect assessed here? The Clo treatment should have depleted the cells, so what was actually measured?

Response:

In our original manuscript, we used nude mice to demonstrate that the antitumor effect of CQ is T cell dependent; then we used Clo-mediated depletion strategy to further demonstrate that the antitumor effect of CQ is macrophage dependent. Since CQ did not

increase T cell function (see the original Supplementary Figure 2a) but increased macrophage function, we speculated that CQ indirectly enhanced T cell function via activating macrophages.

In original Figure 2f and g, we used CQ and Clo to treat B16 melanoma and H22 tumor ascites, respectively. We determined the function of tumor-infiltrating CD8⁺ T cells. We found that the expression of PD1 and CTLA-4 was downregulated in the CD8⁺ T cells after CQ treatment (see the revised Figure 2g,h).

According to the reviewer's comment, **we rewrote the sentence of Figure 2g,h in the revised manuscript, page 7 line 159-162.**

9. *The less acidic pH in lysosomes of M1 macrophages requires a reference.*

Response:

Lysosomal pH value is different between M1 and M2 macrophages. It has been reported that M2 macrophage lysosomes are more acidic than M1 lysosomes (*Mol Biol Cell*.2014;25:3330-41.).

According to the reviewer's suggestion, **we described the lysosomal pH feature of macrophages in the Result section of the revised manuscript, page 11 line 265, and added the reference in the revised manuscript as ref 34, page 33 line 858.**

10. *Line 232, "however" makes no sense here.*

Response:

We deleted this word in revised manuscript.

11. *Throughout the manuscript, some issues with grammar, word use and sentence structure.*

Response:

We thank the reviewer's constructive suggestion. We carefully read the manuscript and corrected any grammar error, improved word use and sentence structure in the revised manuscript.

RESPONSE TO REVIEWER #2:

Reviewer #2 (Remarks to the Author):

The manuscript by Chen et al describes a role for lysosome pH in the regulation of M1 macrophage polarization. The authors show that some of the tumor suppressive effects of treating B16 melanoma or H22 hepatocellularcarcinoma cells in vivo with the lysosome inhibitor, chloroquine is mediated by promoting a switch of tumor associated macrophages from the M2 to M1 tumor killing phenotype. CQ mediated M1-switching is shown to correlate with decreased MDSC and Treg infiltration and increased anti-tumor T cell function in vivo. Mechanistically, the authors show that decreasing lysosome pH following CQ treatment leads to MCOLN1 mediated Ca²⁺ release which correlates with increased p38 and NFκB signaling. In addition, the authors provide evidence that Ca²⁺ release is required for TFEB nuclear translocation and metabolic reprogramming of M1 switched

macrophages.

This study highlights a potentially important function of the lysosome in non-tumor cell populations. It also highlights the potential dual benefit of treating tumors with chloroquine given the potential for combined suppressive effects on the tumor cells as well as infiltrating macrophage populations. However clarification of some comments outlined below, would provide further clarity and novelty to this study.

-The authors only use CQ in all experiments. It would be valuable to show that another lysosome inhibitor – eg. Bafilomycin A1 – or knockdown of a V-ATPase subunit - also has the same effect.

Response:

We thank the reviewer's constructive suggestion. In the revised manuscript, we used bafilomycin A1 (Baf) to treat IL-4-conditioned M2 macrophages. We found that 10nM Baf less increased iNOS, IFN- γ , IL-12p40 expression, compared to CQ. And we increased Baf concentration to 50 nM, which however was toxic and induced apoptosis of macrophages. In addition, we knocked down Atp6v1c1 (a subunit of V-ATPase) in M2 macrophages. Also, the expression of iNOS, IFN- γ , IL-12p40 was weakly upregulated.

According to the reviewer's suggestion, **we added these results in the revised manuscript, page 11 line 271 and revised Supplementary Figure 6b,c,d,e,f.**

-TFEB is a known master regulator of lysosome biogenesis and function. Have the authors checked whether TFEB is responsible for the increase in lysosome biogenesis shown in Fig. 5c? Is there an increase in lamp1/2 staining in these cells?

Response:

In the revised manuscript, we knocked TFEB down in Raw264.7 macrophages and the lysosome numbers were analyzed by lysoTracker. The result showed that TFEB knockdown abrogated CQ-increased lysosome number. Moreover, we stained lamp1 and lamp2 in the macrophages. Consistently, CQ treatment increased lamp1 and lamp2 expression in macrophages, however TFEB knockdown disrupted this process.

According to the reviewer's suggestion, **we added these results in the revised manuscript, page 12 line 281 and revised Figure 5d, page 12 line 289 and revised Supplementary Figure 6h,i,j.**

-What are the relative expression levels of lysosome genes (or CLEAR element genes) in CQ treated macrophages?

Response:

Transcription factor EB (TFEB), a member of the basic helix-loop-helix leucine-zipper family of TFs, is a master regulator of lysosomal function. TFEB direct binds to the consensus sequence (TCACGTGA) at the promoters of lysosomal genes, and such TFEB-binding sequence is named as CLEAR (Coordinated Lysosomal Expression and Regulation) element (*Hum Mol Genet.*2011;20:3852-66.). In the revised manuscript, we found that CQ treatment upregulated the expression of CLEAR gene network (including gene *Arsa*, *Arsb*, *Atp6v0e1*, *C1cn7*, *Ctsa*, *Ctsb*, *Ctsd*, *Ctsf*, *Galns*, *Gba*, *Gla*, *Gns*, *Hexa*, *Lamp1*, *Lamp2*, *Naglu*, *Neu1*, *Psap*, *Scepe1*, *Sgsh*, *Tmem55b*, *Tpp1*) in IL-4-conditioned macrophages.

According to the reviewer's suggestion, we added these results in the revised manuscript, page 14 line 337 and revised Supplementary Figure 9e.

-Does TFEB KD impact lysosome gene expression, number, or pH?

Response:

In the revised manuscript, we knocked TFEB down in Raw264.7 cells, and found that TFEB knockdown inhibited lysosomal gene (*Lamp1* and *Lamp2*) expression and decreased lysosome number in CQ-treated Raw264.7 macrophages. Intriguingly, TFEB knockdown did not affect CQ-increased pH value in macrophages.

According to the reviewer's suggestion, we added these results in the revised manuscript, page 12 line 289 and revised Supplementary Figure 6h,i,j,k.

-KD of TFEB was not able to promote switching from an M2 to M1 state (fig 6g). However KD of TFEB in CQ conditioned macrophages was sufficient to prevent a decrease in infiltrating MDSCs and Treg (fig 7j). How do the authors reconcile these results?

Response:

In this study, we identified that CQ effects on lysosomes and increases the pH value of macrophages, leading to activating Ca^{2+} signaling. Such lysosome-triggered Ca^{2+} signaling activates p38 and NF- κ B for M1 phenotype on one hand, also activates TFEB for macrophage M1 metabolism on the other hand. If we block TFEB with siRNA, it only influences metabolic switch of macrophages but not their phenotype. Since metabolism is the basis for the function exertion, TFEB knockdown may impair macrophage function, leading to prevent CQ-decreased infiltration of MDSCs and Treg cells in tumor microenvironment. Thus, although TFEB has no effect on CQ-reset macrophage phenotype, TFEB knockdown seems to be sufficient to prevent CQ-decreased infiltration of MDSCs and Treg cells in tumor microenvironment via the macrophage-dependent pathway. This inconsistency might be reconciled by that TFEB-regulated glucose metabolism is required for the function exertion of CQ-reset macrophages

According to the reviewer's suggestion, we added this information in the Discussion section of the revised manuscript, page 20 line 488.

-Do the genes shown to be dependent on TFEB regulation (SLC2A1, SLC2A4, PFKM, PKLR) contain TFEB binding elements in their promoter (CLEAR elements)? Can the authors provide CHIP data to show direct regulation?

Response:

In our original manuscript, we used bioinformatical technology (JASPAR database) to predict the TFEB binding site, which indicates that there is consistent element that TFEB binds in the promoter of *Slc2a1*, *Slc2a4*, *Pfkm* and *Pklr*. TFEB knockdown resulted in the downregulation of these genes (see the original Figure 6f).

In the revised manuscript, we further conducted ChIP-qPCR assay. The result showed that TFEB indeed bound to the promoter of these genes.

According to the reviewer's suggestion, we added these results in the revised manuscript, page 14 line 335 and revised Supplementary Figure 9d.

-Nuclear localization is not obvious in panels 5e, l, n. Can the authors provide cytoplasmic vs nuclear fractionation via western blot to show a definitive change in cellular localization upon CQ treatment?

Response:

We appreciate the reviewer's constructive suggestion. In the revised manuscript, we extracted cytoplasmic and nuclear fraction, respectively, in CQ treated macrophages with or without CsA or in Mcoln1-KO Raw264.7 cells. Western blot analysis showed that CQ treatment effectively promoted the entry of TFEB and NF- κ B into the nucleus, however Mcoln1 knockout or CsA pretreatment abrogated the entry of NF- κ B into the nucleus.

According to the reviewer's suggestion, **we added these results in the revised manuscript, page 12 line 287 and revised Supplementary Figure 6g, page 13 line 308 and revised Supplementary Figure 7e,f.**

-Level of IL-10 (M2) secretion in fig 3g could also be included.

Response:

We thank the reviewer's constructive suggestion. In the revised manuscript, we also analyzed the levels of IL-10, and the result showed that CQ treatment resulted in the decreased of IL-10 in tumor tissue.

According to the reviewer's suggestion, **we added these results in the revised manuscript, page 9 line 214 and revised Figure 3i.**

-some spelling and grammatical errors were found in the text.

Response:

We thank the reviewer for this constructive suggestion. We carefully read the manuscript and corrected any spelling and grammatical errors in the revised manuscript.

RESPONSE TO REVIEWER #3:

Reviewer #3 (Remarks to the Author):

The manuscript by Chen et al, describes a novel anti-tumoral action of chloroquine (CQ) that entails the switching of TAMs from M2 to a M1 tumor-killing phenotype supporting anti-tumor immunity responses, by reducing infiltration of immunosuppressive Tregs and MDSCs. The authors analyse the mechanisms of the effect of CQ on TAMs and find that it involves a Ca²⁺-mediated TFEB-dependent pathway reprogramming TAM metabolism. The study is relevant for our understanding of the in vivo effects of CQ. However, several aspects of this study remains to be further validated experimentally in order to support author's conclusions.

General remarks:

In all ex vivo or in vitro experiments with immune cells that the authors have presented, the authors have completely excluded the role of cancer cells. CQ can exert cytotoxic and immunosuppressive effects on or via cancer cells and these effects also play a role in immunological modulation. Considering that a tumor is usually >90% cancer cells, the

authors cannot exclude the immunological effects of in their in vitro or in vivo experiments. Thus the immune cell co-culture experiments with DCs or macrophages presented in this manuscript should be repeated in presence of cancer cells as well (B16 and hepatocellular carcinoma) in order to rule in or rule out the contributory or inhibitory roles of cancer cells.

Response:

We thank the reviewer's insightful suggestion. In the revised manuscript, we treated B16 or H22 with 10 μ M CQ for 24 hours, then the treated B16 or H22 cells were collected and co-cultured with macrophages or DCs at a 10:1 ratio (more tumor cell and less immune cells). We found that neither CQ-untreated nor CQ-treated tumor cells upregulated the expression of CD80, CD86, MHC II and CCR7, by contrast, these tumor cells enhanced arginase1 and IL-10 expression in macrophages. These data suggest that the CQ-effected tumor cells are not able to promote macrophage polarization to M1 phenotype.

According to the reviewer's suggestion, we added these results in the revised manuscript, page 6 line 140 and revised Supplementary Figure 2d, page 9 line 204 and revised Supplementary Figure 4g.

The authors argue that CQ enhances tumor killing by resetting TAMs towards M1 but they don't provide any data on immune-mediated tumor cell killing. For macrophages, simply showing increase in iNOS levels does not mean that the macrophages are killing cancer cells via respiratory burst. This actually needs to be shown (cancer cell death in presence of CQ+macrophages or only macrophages and only CQ, plus, NO levels in media to start with) and iNOS or NO activity should be blocked to establish this. Also T cell-based cancer cell killing analysis should be performed. The authors moreover present no analysis on perforin-granzyme positivity of T cells in vivo, or whether T cells derived from CQ-treated tumors can kill B16 cells ex vivo/ in vitro. All these assays are necessary to validate conclusions of the authors (including the title of this paper).

Response:

We appreciate the reviewer's constructive suggestion. In the revised manuscript, we measured NO levels after CQ treatment and found that CQ treatment resulted in the increased NO levels in IL-4-conditioned M2 macrophages. We then co-cultured tumor cells with CQ-treated or untreated M2 macrophages. We found that CQ-treated M2 macrophages only slightly caused tumor cell death, and inhibition of NO production resulted in the disruption of such slight killing, suggesting that CQ-polarized macrophages may directly induce a small scale tumor cell death in a NO-dependent manner, especially considering that T cell depletion leads to abrogating CQ-mediated antitumor effect (see original Figure 1f,g,h,i). Then, we clarified the role of T cell immunity in CQ-mediated antitumor effect. OVA-B16 melanoma-bearing CD45.1⁺ C57BL/6 mice were treated with or without CQ, concomitant with adoptive transfer of OVA-specific CD8⁺ T cells from CD45.2⁺ OT-I transgenic mice. We found that CQ treatment enhanced tumor growth inhibition mediated by CD8⁺ T cells, which however was blocked by macrophage depletion. In line with this result, upon CQ treatment, tumor-infiltrating CD45.2⁺ T cells upregulated IFN- γ evidenced by flow cytometry as well as upregulated granzyme B in OVA-B16 tumor tissues evidenced by immunohistochemical staining. In addition, we isolated tumor-infiltrating CD45.2⁺ cells and co-cultured with OVA-B16 cells at a ratio of 30:1 for cytotoxicity assay.

The result showed that tumor-specific T cells from CQ-treated mice more efficiently killed OVA-B16 cells, compared to T cells from untreated mice. Together, these data suggest that CQ seems not to mobilize M1 macrophages polarized as the major killing arm against tumor but mobilizes polarized macrophages to remodel the antitumor microenvironment, thus facilitating the activation of tumor-specific T cells.

According to the reviewer's suggestion, **we added these results in the revised manuscript, page 8 line 188 and revised Figure 3c, page 8 line 197 and revised Supplementary Figure 4f, page 7 line 167 and revised Figure 2i and Supplementary Figure 3d,e,f.**

Specific points:

Figure 1. There is a discrepancy in the data comparing immunocompetent mice with immunodeficient ones. Why are immunocompetent mice always treated with higher dose of CQ (75mg/kg) whereas immunodeficient mice or nude mice are always treated with lower dose (50 mg/kg) of CQ? Similar doses of CQ should be used in both mice cohorts in order to rule out that differences in tumor growth are because of different doses of CQ rather than T cell dependency.

Response:

We thank the reviewer's constructive comments. In the revised manuscript, we used high dosage of CQ (75mg/kg) to treat nude mice bearing B16 melanoma or H22 hepatocarcinoma. The result was similar to that from 50 mg/kg treatment.

According to the reviewer's suggestion, **we revised Figure 1f,g.**

Figure 1. CD3 depletions assays need more statistical analysis to be valid. No statistics are shown for CQ+CD3 condition vs CQ alone. Is the effect of CD3 antibodies even significant? CQ only conditions has no error bars in some places? That is improbable.

Also no interpretation provided for why CD3 depletion has different effect on PBS and CQ conditions?

Response:

In the original manuscript, we used anti-mouse CD3 ϵ (clone 145-2C11) to deplete CD3⁺ T cells with a protocol of 200 μ g/mouse every 4 days via i.p. injection. T cell depletion with such protocol counteracted CQ-mediated antitumor effect (see original Fig. 1h). However, such protocol might not completely deplete CD3⁺ T cells, regarding to once depletion per 4 days. Notably, this antibody also has agonist effect that activates CD3⁺ T cells (*Immunity.2014;41:555-66; Immunity.2015;42:68-79.*). Such T cell activation might result in the inhibition of tumor growth (*Science.1988;242:569-71.*), as shown between the PBS group and CD3 depletion group in original Figure 1h. To verify this, in the revised manuscript, we additionally intraperitoneally injected 200 μ g anti-CD3 Ab to tumor-bearing mice once another day to better deplete T cells. Under such condition, we found that T cell depletion accelerated tumor growth, compared to PBS group. Since the purpose of this figure was to further demonstrate T cell-dependent antitumor effect of CQ, we removed the PBS control from the original Figure 1h in order to make the result clearer and avoid any confusion. Also, we added all the statistics in the revised Figure 1h and i. In addition, we diminished the connecting line to better present the error bars.

According to the reviewer's suggestion, **we revised Figure 1h and Supplementary Figure**

1b.

The assay with splenic T cells that the authors carry out ex vivo to rule out direct effect of CQ on T cells is not representative of this effect. Splenic T cells should be derived from the immunocompetent mice treated in Fig 1 with CQ (75mg/kg) and these should be analyzed for markers of cell death (Annexin V-PI), activation (CD69, LAMP1, CD137) and T cell anergy (PD1, CTLA4).

Response:

We thank the reviewer's constructive suggestion. In the original manuscript, we isolated and treated splenic T cells with anti-CD3 and CD28 antibodies in the presence or absence of CQ. We did not observe the promoting effect of CQ on T cell proliferation. In the revised manuscript, we further isolated splenic T cells from CQ (75 mg/kg) treated and untreated mice, respectively. Consistently, T cells in CQ treated mice did not show difference from that in untreated mice, including T cell death (Annexin-V and PI staining), T cell activation (CD69, CD137 marker), lysosome status (Lamp1) as well as T cell anergy (PD1, CTLA-4). According to the reviewer's suggestion, **we added these results in the revised manuscript, page 6 line 130 and revised Supplementary Figure 2b.**

Fig 2. The T cell immunophenotyping is not correctly done. CD8+IFN γ + cells include both CD8+T cells and NK cells - the authors need to quantify CD45+CD3+CD8+IFN γ + population for CD8+T cells. Moreover it is not clear what is the physiological relevance of results in Fig 2d,e. that is - 4% increase in CD8+IFN γ + cells is not really a huge amount for enforcing CD8+T cell-driven immunity. The authors should better deplete CD8+ cells with antibodies to confirm the relevance of this population in this system. Also it is not clear what population is labelled with CTLA4 or PD1? More markers should be used for T cells, macrophages/myeloid cells and cancer cells to differentiate between different tumoral populations expressing these immune-checkpoints.

The correlation between IFN γ and PD1/CTLA4 levels makes no sense. Sustained IFN γ production up-regulates PD1/CTLA4 as a negative feedback loop mechanism, this is the whole basis of the concept of immune-checkpoints, whereas the authors see the opposite of this. This data needs to be clarified better. More later time-points should be added and direct effects of CQ on PD1/CTLA4 should be better presented.

Response:

CD8 molecule recognizes and binds MHC class I α chain, which is expressed in CD8⁺ T cells and also expressed in some DCs and NK cells. In our original manuscript, we actually gated the CD3⁺ cells and then analyzed CD8⁺IFN- γ ⁺ cells (please see the figure legend of Figure 2d,e in the original manuscript, page 33 line 813-819). For Figure 2d, e, we isolated immune cells from tumor tissues and directly analyzed CD3-gated CD8⁺IFN- γ ⁺ cells by flow cytometry without activation of these cells *in vitro* as well as addition of Golgi inhibitor. In this case, IFN- γ is able to be released to the extracellular space and the intracellular detection of IFN- γ is low.

In the revised manuscript, we isolated the tumor-infiltrating immune cells and treated them with anti-CD3 and CD28 antibodies *in vitro* for 24 hours. Monensin was added for the last 6 hours to block Golgi transport function. Upon this *in vitro* stimulation, we found that more

than 20% CD3⁺CD8⁺ T cells are IFN- γ positive. We used this higher percentage of IFN- γ ⁺ in CD3⁺CD8⁺ T cells to replace the original one (original Figure 2d,e). We also conducted CD8 T cell depletion experiment to further demonstrate that CQ treatment causes IFN- γ production by CD8 T cells. We found that CD8 T cell depletion with anti-CD8 depleting antibody resulted in the loss of most IFN- γ in tumor tissue.

According to the reviewer's suggestion, **we revised Figure 2d,e and added new results in the revised manuscript, page 7 line 156 and revised Figure 2f.**

PD-1 and CTLA-4 are two important inhibitory molecules expressed on T cell membrane surface. Although IFN- γ is capable of upregulating PD-L1 and IDO1 to induce negative immunoregulation, it does not regulate PD-1 expression. To date, the mechanism of PD-1 upregulation in tumor-infiltrating T cells remains unclear. On the other hand, IFN- γ generally does not regulate CTLA-4 expression. CTLA-4 is commonly expressed on T cell membrane, which however is internalized upon tyrosine phosphorylation of cytoplasmic tail (*J Cell Biochem.*2000;78:241-50.) and can be recycled to the membrane or undergo degradation upon signals T cells receive (*J Biol Chem.*2012;287:9429-40.). In this study, we observed that CQ treatment resulted in the upregulation of IFN- γ and downregulation of PD1/CTLA-4 in tumor-infiltrating CD8⁺ T cells. To further verify this, we additionally analyzed T-bet, an essential transcription factor for IFN- γ production in CD8⁺ T cells (*Science.*2002;295:338-42; *Proc. Natl. Acad. Sci. USA.*2002;99:5545-50.). We found that T-bet was upregulated in tumor-infiltrating CD3⁺CD8⁺ T cells upon CQ treatment, while the expression of PD1/CTLA-4 was downregulated in CD3⁺CD8⁺T-bet⁺ T cells, consistent with previous report that T-bet represses PD-1 expression (*Nat Immunol.*2011;12:663-71.).

According to the reviewer's suggestion, **we added these results in the revised manuscript, page 7 line 162 and revised Supplementary Figure 3c.**

IL12p40 analysis is not a correct marker for macrophage activation. IL12p70 is the actual active subunit of IL12. While IL12p40 is generally easily up-regulated, in absence of IL12p70, IL12p40 cannot enforce any immunological activity. Analysis for IL12p70 should also be provided and to support author's conclusions, it should follow the same trend as IL12p40.

Response:

We thank the reviewer's constructive suggestion. In the revised manuscript, we measured IL-12p70 in CQ-treated macrophages both mouse and human by ELISA. In line with IL-12p40, IL-12p70 production was elevated upon CQ treatment.

According to the reviewer's suggestion, **we added these results in the revised manuscript, page 8 line 189 and revised Supplementary Figure 4c, page 9 line 222 and revised Supplementary Figure4j.**

The authors have analyzed the level of immunostimulatory M1 cytokines but not those of immunosuppressive M2 cytokines. IL10, TGF β and IDO1 levels should also be analyzed in media of M1 and M2 assays to clarify functional M2 polarisations and address the possibilities of partial functional suppression within M1 macrophages.

Response:

In the revised manuscript, we additionally analyzed IL-10, TGF- β 1 and IDO1 in CQ-treated

M2 macrophages by ELISA (IL-10, TGF- β 1) or real-time PCR (*IL-10*, *TGF- β 1*, *IDO1*). We found that the expression of IL-10, TGF- β 1 and IDO1 was downregulated with CQ treatment.

According to the reviewer's suggestion, we added these results in the revised manuscript, page 8 line 195 and revised Supplementary Figure 4e.

Fig 3. 10-15% positivity of total macrophages for respective cytokines isn't very physiologically high. The authors need to show actual concentrations of these cytokines in the medium and these concentrations need to (significantly) exceed 10 pg/mL to be physiologically relevant.

Response:

In the revised manuscript, we detected the concentration of IFN- γ , IL-12p70 and TNF- α by ELISA. The result showed that CQ treatment elevated these cytokines' concentration to 100-800pg/mL.

According to the reviewer's suggestion, we added these results in the revised manuscript, page 8 line 189 and revised Supplementary Figure 4c.

Fig 3. Why are macrophages in Fig 3b already more than 60% positive for CD80? This means that the macrophages are pre-stimulated on a certain level. The authors need to optimize the macrophage cultures to have CD80 positivity below 20%.

Response:

In the original manuscript, we pre-incubate macrophages on ice with anti-CD16/CD32 (clone 93 mAb) to block non-specific binding of immunoglobulin to macrophage Fc receptors. We then conducted flow cytometric analysis. The high proportion of CD80 might be due to that the CD16/CD32 Ab lost its efficacy. In the revised manuscript, we used a new one and repeated the assay again. The result showed a consistent trend but the CD80 positive proportion became 15%.

According to the reviewer's suggestion, we revised Figure 3b,d.

Fig 7. CD11b+Gr1(hi) are not a specific markers for MDSCs. These markers mark all neutrophils and granulocytes. In order to have a valid MDSC panel the authors need to have CD11b+GR1(hi)CD86(lo)MHCII(lo) wherein low expression of CD86 and MHCII is marker of the possible suppressive nature of these granulocytes. Also MDSCs, are not just granulocytic in origin but also myeloid-MDSCs need to be analyzed by a similar strategy. The authors need to deplete MDSCs in vivo to prove their conclusions regarding MDSCs.

Response:

Accumulation of immature myeloid cells and immune suppression was recognized in both tumor-bearing mice and cancer patients in 1980s (*Cancer Res.*1987;1:100-5; *J Immunol.*1989;2:491-8.). Because of the suppressive activities and heterogeneity of these myeloid cells, the term myeloid-derived suppressor cell (MDSC) was proposed to denote this population (*Cancer Res.*2007;1:425.). MDSCs potentially differentiate toward monocyte/macrophages or granulocytes. Correspondingly, MDSCs in mouse can be classified as monocytic MDSCs (CD11b⁺Ly6C^{hi}Ly6G⁻) and granulocytic MDSCs (CD11b⁺Ly6C^{lo}Ly6G⁺) and both of them favor tumor growth in many tumor models (*Cancer*

Cell.2017;32:654-68; Cancer Discov.2017;1:72-85; Cancer Cell.2016;30:108-19; Trends Immunol.2016;37:208-20.) CD11b⁺Gr-1⁺ MDSCs actually contain these two subsets, since the anti-Gr-1 antibody (RB6-8C5) recognizes both Ly6C and Ly6G. Thus, in the original manuscript, we did not further clarify which subsets of MDSCs in mouse tumor model was influenced by CQ treatment. In the revised manuscript, we further analyzed CD11b⁺Ly6C^{hi}Ly6G⁻ monocytic and CD11b⁺Ly6C^{lo}Ly6G⁺ granulocytic MDSCs. We found that CQ treatment actually decreased both subsets of MDSCs. Such CQ-mediated decrease could be disrupted by macrophage depletion, Mcoln1 Knockout or TFEB knockdown.

According to the reviewer's suggestion, we added these results in the revised manuscript, page 15 line 366 and revised Supplementary Figure 10a,b,c,d,f.

We previously did a lot of work on MDSCs and used Gr-1 antibody to efficiently deplete MDSCs (*Cancer Res.2006;66:1123-31; Cancer Lett.2007;252:86-92; PLoS One.2010;5:e8922; J Immunol.2010;185:7199-206.*). In the revised manuscript, we further used anti-Gr-1 depleting antibody (BioLegend) to deplete both granulocytic and monocytic MDSCs. Under this condition, CQ-mediated antitumor effect was impaired in B16 melanoma tumor model.

According to the reviewer's suggestion, we added these results in the revised manuscript, page 16 line 386 and revised Supplementary Figure 10j.

Cyclophosphamide is not a selective Treg depleting methodology, it has several other effects on the tumor (e.g. immunogenic cell death) and gut microbiota (Th17 anti-tumor axes). These data should be replaced by Tregs depleted in the current system via anti-CD25 antibodies. Finally chlodronate liposome-based depletion is not specific for macrophages only but targets all phagocytes.

Response:

Cyclophosphamide (CY) is a prominent alkylating anticancer agent that induces tumor cells into death. Low-dose CY, although not directly killing tumor cells, is able to activate antitumor immunity by selective depletion of Tregs, leading to therapeutic effect against tumors. We previously elucidated the underlying mechanism through which low-dose CY selectively deplete Treg cells but not conventional CD4 T cells (*Cancer Res.2010;70:4850-8.*); we further found that 25 mg/kg dosage of CY is optimal for Treg depletion in mouse tumor model (*J Immunol.2010;185:7199-206.*). With high dosage (such as 100 mg/kg) of CY treatment, tumor cells can be killed and release tumor antigens and danger signals, leading to immunogenic cell death (*Cancer Cell.2015;28:690-714.*). Also, under high dosage (100 mg/kg) condition, CY destroyed gut integrity, leading to Th17 antitumor axes (*Science.2013;342:971-6.*). However, in this study, we used low-dose CY (25 mg/kg), which did not result in immunogenic cell death nor activated Th17 antitumor axes. In the revised manuscript, we additionally used anti-CD25 depleting antibody (PC-61), a widely used method to deplete Treg cells in mouse model (*Nature.2016;530:434-40; Nature Commun.2015;6:7566; Leukemia.2015;29:947-57.*). Consistent with low dose CY effect, CD25 depletion also enhanced the inhibitory effect of CQ on B16 or H22 tumor growth.

According to the reviewer's suggestion, we added these results in the revised manuscript,

page 16 line 384 and revised Supplementary Figure 10h,i.

Macrophages are critical innate immune cell subset with high heterogeneity. The completely specific marker for macrophages is not identified. There always are cross-marker(s) and cross-function(s) between macrophages and other immune cells. Currently, there is no specific agent to target macrophages. Clodronate liposome since its discovery is widely used to deplete macrophages in various tumor models as a relatively specific agent, and their feasibility and efficiency have also been well demonstrated (*Nat Commun.*2016;7:11051; *Cancer Cell.*2014;26:190-206; *Nature Medicine.*2013;19:1166-72; *Nature Medicine.*2013;19:429-36; *Nature Immunology.*2013;14:574-83.).

Reviewer #1 (Remarks to the Author):

All technical and statistical concerns have been addressed.

Reviewer #2 (Remarks to the Author):

The authors have sufficiently addressed my questions and recommendations for improving the manuscript. A significant amount of additional data has been added that further strengthen the claims of the study. Overall the findings by Chen, Xie et al will be broadly applicable to the community and warrant acceptance for publication in Nature Communications.

Reviewer #3 (Remarks to the Author):

I am happy to see that the authors have taken into serious consideration most of the comments raised by this reviewer and performed a battery of new experiments supporting their conclusions. The manuscript has been largely improved and several inconsistencies have been solved. I would nevertheless pinpoint that some of the additional sentences need to be more carefully checked for grammatical mistakes.

Reviewers' comments:

Reviewer #1 (Remarks to the Author):

The authors used chloroquine (CQ) to repolarize tumor macrophages to an M1 phenotype. This is an interesting approach, although not entirely new (see ref. 3, which should be more clearly acknowledged as a previous paper applying the same concept, and other earlier work). As a mechanism, the authors propose that CQ induces calcium release from lysosomes, which repolarizes macrophage metabolism in several ways: (1) calcium release through mucolipin-1, (2) nuclear translocation of TFEB resulting in (3) increased glycolysis as demonstrated by Seahorse, (4) p38 phosphorylation resulting in (5) nuclear translocation of NFkB, a known M1 polarizer. Thus, there is a convincing pathway from CQ effects causing less acidic lysosomes to the M1 polarization.

1. The authors propose that chloroquine is better than macrophage depletion by clodronate liposomes. This comparison (CQ vs Clo) must be tested statistically, correcting for multiple comparisons. If the difference is not significant, the overall concept is of questionable importance for cancer.
2. Human macrophages do not express much iNOS, so other markers are commonly used for M1. The key in vitro experiments should be repeated with human macrophages, and their polarization assessed. This is important for clinical translation.
3. Figure 1c is underpowered (only 6 mice per group).
4. Figure 2 shows some reduction in PD1 and CTLA4 expression, but both reductions are not complete. Therefore, the chloroquine should be tested in combination with the inhibitor of PI3Kgamma, a known M1 polarizer (ref. 3). In the same vein, only 15-20% of TAMs express IL-12 or IFN γ after CQ treatment. What about the others? Do they at least lose arginase?
5. Figure 4f requires a low-power overview to appreciate how many cells switch on NFkB
6. Figure 5l is lacking untreated and scrambled controls.
7. How does 10 μ M CQ (the in vitro dose) correspond to 75 mg/kg (the clinical dose)?
8. Figure 2f and g: How was the CQ effect assessed here? The Clo treatment should have depleted the cells, so what was actually measured?
9. The less acidic pH in lysosomes of M1 macrophages requires a reference.
10. Line 232, "however" makes no sense here
11. Throughout the manuscript, some issues with grammar, word use and sentence structure.

Reviewer #2 (Remarks to the Author):

The manuscript by Chen et al describes a role for lysosome pH in the regulation of M1 macrophage polarization. The authors show that some of the tumor suppressive effects of treating B16 melanoma or H22 hepatocellular carcinoma cells in vivo with the lysosome inhibitor, chloroquine is mediated by promoting a switch of tumor associated macrophages from the M2 to M1 tumor killing phenotype. CQ mediated M1-switching is shown to correlate with decreased MDSC and Treg infiltration and increased anti-tumor T cell function in vivo. Mechanistically, the authors show that decreasing lysosome pH following CQ treatment leads to MCOLN1 mediated Ca²⁺ release which correlates with increased p38 and NFκB signaling. In addition, the authors provide evidence that Ca²⁺ release is required for TFEB nuclear translocation and metabolic reprogramming of M1 switched macrophages.

This study highlights a potentially important function of the lysosome in non-tumor cell populations. It also highlights the potential dual benefit of treating tumors with chloroquine given the potential for combined suppressive effects on the tumor cells as well as infiltrating macrophage populations. However clarification of some comments outlined below, would provide further clarity and novelty to this study.

-The authors only use CQ in all experiments. It would be valuable to show that another lysosome inhibitor – eg. Bafilomycin A1 – or knockdown of a V-ATPase subunit - also has the same effect.

-TFEB is a known master regulator of lysosome biogenesis and function. Have the authors checked whether TFEB is responsible for the increase in lysosome biogenesis shown in Fig. 5c? Is there an increase in lamp1/2 staining in these cells?

-What are the relative expression levels of lysosome genes (or CLEAR element genes) in CQ treated macrophages?

-Does TFEB KD impact lysosome gene expression, number, or pH?

-KD of TFEB was not able to promote switching from an M2 to M1 state (fig 6g). However KD of TFEB in CQ conditioned macrophages was sufficient to prevent a decrease in infiltrating MDSCs and Treg (fig 7j). How do the authors reconcile these results?

-Do the genes shown to be dependent on TFEB regulation (SLC2A1, SLC2A4, PFKM, PKLR) contain TFEB binding elements in their promoter (CLEAR elements)? Can the authors provide CHIP data to show direct regulation?

-Nuclear localization is not obvious in panels 5e, l, n. Can the authors provide cytoplasmic vs nuclear fractionation via western blot to show a definitive change in cellular localization upon CQ treatment?

-Level of IL-10 (M2) secretion in fig 3g could also be included.

-some spelling and grammatical errors were found in the text.

Reviewer #3 (Remarks to the Author):

The manuscript by Chen et al, describes a novel anti-tumoral action of chloroquine (CQ) that entails the switching of TAMs from M2 to a M1 tumor-killing phenotype supporting anti-tumor immunity responses, by reducing infiltration of immunosuppressive Tregs and MDSCs. The authors analyse the mechanisms of the effect of CQ on TAMs and find that it involves a Ca²⁺-mediated TFEB-dependent pathway reprogramming TAM metabolism. The study is relevant for our understanding of the in vivo effects of CQ. However, several aspects of this study remains to be further validated experimentally in order to support author's conclusions.

General remarks:

In all ex vivo or in vitro experiments with immune cells that the authors have presented, the authors have completely excluded the role of cancer cells. CQ can exert cytotoxic and immunosuppressive effects on or via cancer cells and these effects also play a role in immunological modulation. Considering that a tumor is usually >90% cancer cells, the authors cannot exclude the immunological effects of CQ-treated cancer cells in their in vitro or in vivo experiments. Thus the immune cell co-culture experiments with DCs or macrophages presented in this manuscript should be repeated in presence of cancer cells as well (B16 and hepatocellular carcinoma) in order to rule in or rule out the contributory or inhibitory roles of cancer cells.

The authors argue that CQ enhances tumor killing by resetting TAMs towards M1 but they don't provide any data on immune-mediated tumor cell killing. For macrophages, simply showing increase in iNOS levels does not mean that the macrophages are killing cancer cells via respiratory burst. This actually needs to be shown (cancer cell death in presence of CQ+macrophages or only macrophages and only CQ, plus, NO levels in media to start with) and iNOS or NO activity should be blocked to establish this. Also T cell-based cancer cell killing analysis should be performed. The authors moreover present no analysis on perforin-granzyme positivity of T cells in vivo, or whether T cells derived from CQ-treated tumors can kill B16 cells ex vivo/ in vitro. All these assays are necessary to validate conclusions of the authors (including the title of this paper).

Specific points:

Figure 1. There is a discrepancy in the data comparing immunocompetent mice with

immunodeficient ones. Why are immunocompetent mice always treated with higher dose of CQ (75mg/kg) whereas immunodeficient mice or nude mice are always treated with lower dose (50 mg/kg) of CQ? Similar doses of CQ should be used in both mice cohorts in order to rule out that differences in tumor growth are because of different doses of CQ rather than T cell dependency.

Figure 1. CD3 depletions assays need more statistical analysis to be valid. No statistics are shown for CQ+CD3 condition vs CQ alone. Is the effect of CD3 antibodies even significant? CQ only conditions has no error bars in some places? That is improbable. Also no interpretation provided for why CD3 depletion has different effect on PBS and CQ conditions?

The assay with splenic T cells that the authors carry out ex vivo to rule out direct effect of CQ on T cells is not representative of this effect. Splenic T cells should be derived from the immunocompetent mice treated in Fig 1 with CQ (75mg/kg) and these should be analyzed for markers of cell death (Annexin V-PI), activation (CD69, LAMP1, CD137) and T cell anergy (PD1, CTLA4).

Fig 2. The T cell immunophenotyping is not correctly done. CD8+IFN γ + cells include both CD8+T cells and NK cells - the authors need to quantify CD45+CD3+CD8+IFN γ + population for CD8+T cells. Moreover it is not clear what is the physiological relevance of results in Fig 2d,e. that is - 4% increase in CD8+IFN γ + cells is not really a huge amount for enforcing CD8+T cell-driven immunity. The authors should better deplete CD8+ cells with antibodies to confirm the relevance of this population in this system. Also it is not clear what population is labelled with CTLA4 or PD1? More markers should be used for T cells, macrophages/myeloid cells and cancer cells to differentiate between different tumoral populations expressing these immune-checkpoints.

The correlation between IFN γ and PD1/CTLA4 levels makes no sense. Sustained IFN γ production up-regulates PD1/CTLA4 as a negative feedback loop mechanism, this is the whole basis of the concept of immune-checkpoints, whereas the authors see the opposite of this. This data needs to be clarified better. More later time-points should be added and direct effects of CQ on PD1/CTLA4 should be better presented.

IL12p40 analysis is not a correct marker for macrophage activation. IL12p70 is the actual active subunit of IL12. While IL12p40 is generally easily up-regulated, in absence of IL12p70, IL12p40 cannot enforce any immunological activity. Analysis for IL12p70 should also be provided and to support author's conclusions, it should follow the same trend as IL12p40.

The authors have analyzed the level of immunostimulatory M1 cytokines but not those of immunosuppressive M2 cytokines. IL10, TGF β and IDO1 levels should also be analyzed in media of M1 and M2 assays to clarify functional M2 polarisations and

address the possibilities of partial functional suppression within M1 macrophages.

Fig 3. 10-15% positivity of total macrophages for respective cytokines isn't very physiologically high. The authors need to show actual concentrations of these cytokines in the medium and these concentrations need to (significantly) exceed 10 pg/mL to be physiologically relevant.

Fig 3. Why are macrophages in Fig 3b already more than 60% positive for CD80? This means that the macrophages are pre-stimulated on a certain level. The authors need to optimize the macrophage cultures to have CD80 positivity below 20%.

Fig 7. CD11b+Gr1(hi) are not a specific markers for MDSCs. These markers mark all neutrophils and granulocytes. In order to have a valid MDSC panel the authors need to have CD11b+GR1(hi)CD86(lo)MHCII(lo) wherein low expression of CD86 and MHCII is marker of the possible suppressive nature of these granulocytes. Also MDSCs, are not just granulocytic in origin but also myeloid-MDSCs need to be analyzed by a similar strategy. The authors need to deplete MDSCs in vivo to prove their conclusions regarding MDSCs.

Cyclophosphamide is not a selective Treg depleting methodology, it has several other effects on the tumor (e.g. immunogenic cell death) and gut microbiota (Th17 anti-tumor axes). These data should be replaced by Tregs depleted in the current system via anti-CD25 antibodies. Finally clodronate liposome-based depletion is not specific for macrophages only but targets all phagocytes.

RESPONSES TO REVIEWERS

We would like to express our sincere thanks to all three reviewers for their critical and constructive comments. We have performed substantial additional experiments to address their concerns. We respond point-by-point to each of their comments and criticisms. We feel that their comments have helped us on significantly improving and strengthening the manuscript and clarifying some issues. We hope that the revision has addressed their major concerns.

RESPONSE TO REVIEWER #1:

Reviewer #1 (Remarks to the Author):

The authors used chloroquine (CQ) to repolarize tumor macrophages to an M1 phenotype. This is an interesting approach, although not entirely new (see ref. 3, which should be more clearly acknowledged as a previous paper applying the same concept, and other earlier work). As a mechanism, the authors propose that CQ induces calcium release from lysosomes, which repolarizes macrophage metabolism in several ways: (1) calcium release through mucolipin-1, (2) nuclear translocation of TFEB resulting in (3) increased glycolysis as demonstrated by Seahorse, (4) p38 phosphorylation resulting in (5) nuclear translocation of NFkB, a known M1 polarizer. Thus, there is a convincing pathway from CQ effects causing less acidic lysosomes to the M1 polarization.

1. The authors propose that chloroquine is better than macrophage depletion by clodronate liposomes. This comparison (CQ vs Clo) must be tested statistically, correcting for multiple comparisons. If the difference is not significant, the overall concept is of questionable importance for cancer.

Response:

We thank the reviewer's constructive comment. CQ is an alkaline agent that can increase lysosomal pH value of M2 macrophages. As a result, CQ does not deplete M2 macrophages but instead reset them into M1 phenotype. In this regard, transition of M2 to M1 not only relieves the tumor-promoting effect (M2 depletion) but also acquires new tumor-inhibiting effect (M1 generation), thus theoretically better than single macrophage depletion.

In our original data, we showed that macrophages depletion led to the disruption of the inhibitory effect of CQ on B16 melanoma as well as H22 hepatocarcinoma (Fig. 2a-c). In the revised manuscript, we compared the

statistics between the CQ and Clo two groups. We found that the effect of CQ was statistically better than the effect of Clo on B16 melanoma (Fig. 2a) and on H22 hepatocarcinoma ascites (Fig. 2c), suggesting that CQ-based strategy is better than macrophage depletion. In addition, in H22 muscle hepatocarcinoma model, we did not observe the statistical difference between CQ and Clo (Fig. 2b). This might be due to the tissue specificity. The muscle tissue intrinsically produces abundant lactic acid, and the latter effectively polarizes macrophages to an M2-like state (*Nature.2014;513:559-63.*). Thus, even if CQ effectively educates M2 to M1 in H22 muscle hepatocarcinoma, the endogenous lactic acid may impair this process by re-educating M1 back to M2.

According to the reviewer's suggestion, we added the statistical asterisk in the revised Figure 2a and 2c.

2. Human macrophages do not express much iNOS, so other markers are commonly used for M1. The key in vitro experiments should be repeated with human macrophages, and their polarization assessed. This is important for clinical translation.

Response:

In the revised manuscript, we isolated monocytes from healthy donors' blood and cultured them with 20 ng/mL human M-CSF to induce the differentiation toward macrophages. Seven days later, we added 20 ng/mL human IL-4 for M2-like polarization. Then, we used CQ to treat such human M2-like macrophages. These treated cells were analyzed by flow cytometry (CD163, CD206, CD64, IFN- γ , TNF- α , TGF- β 1, IL-10) and real-time qPCR (*Arg1, IFN- γ , IL-12p40, TNF- α , TGF- β 1, IL-10*), and the supernatants were used for ELISA (IFN- γ , TNF- α , IL-12p40, IL-12p70, TGF- β 1, IL-10). Consistently, CQ treatment also induced human M2 macrophages transiting to M1, as evidenced by upregulation of IFN- γ , TNF- α , IL-12p70, but downregulation of CD163, CD206, CD64, TGF- β 1 and IL-10.

According to the reviewer's suggestion, we added these results in the revised manuscript, page 9 line 219 and revised Supplementary Figure 4h,i,j.

3. Figure 1c is underpowered (only 6 mice per group).

Response:

We repeated the B16 lung metastasis experiment with CQ treatment. Eight mice were additional used in each group. Consistent with our original data, CQ markedly decreased the number of tumor nodules in the lungs.

According to the reviewer's suggestion, we revised Figure 1c (n=14).

4. Figure 2 shows some reduction in PD1 and CTLA4 expression, but both reductions are not complete. Therefore, the chloroquine should be tested in combination with the inhibitor of PI3Kgamma, a known M1 polarizer (ref. 3). In the same vein, only 15-20% of TAMs express IL-12 or IFN γ after CQ treatment.

What about the others? Do they at least lose arginase?

Response:

PI3K γ has been shown to be highly expressed in tumor-associated myeloid cells and inhibition of PI3K γ activity by IPI-549 restores the sensitivity to immune checkpoint blocking therapy in mouse tumor model (*Nature.2016;539:443-7.*). Here, we combined CQ with PI3K γ inhibitor IPI-549 to treat B16 melanoma and H22 hepatocarcinoma. We found that the inhibitory effect of CQ on the growth of B16 melanoma and H22 hepatocarcinoma was enhanced by IPI-549, concomitant with the further decreased PD1 and CTLA-4 expression, suggesting that PI3K γ inhibitor has a synergistic effect on CQ-triggered antitumor immunity.

According to the reviewer's suggestion, we added these results in the revised manuscript, page 9 line 223 and revised Supplementary Figure 4k,l,m.

In our original Fig. 3a, we used CQ to treat IL-4-conditioned M2 macrophages, and found a significant upregulation of M1-related cytokine expression (IFN- γ , IL-12p40 and TNF- α). The proportion of these expressing cells is around 15-20%. This might be due to that conversion of M2 to M1 is a slow process and the macrophages were at different stages. In the revised manuscript, we used APC to label IL-12p40, IFN- γ and used PE to label arginase1, and found that CQ treatment resulted in 30% arginase1⁺ converting to arginase1⁻ among IL-12⁻ and IFN- γ ⁻ macrophages.

According to the reviewer's suggestion, we added these results in the revised manuscript, page 9 line 201 and revised Figure 3f.

5. Figure 4f requires a low-power overview to appreciate how many cells switch on NF κ B.

Response:

In the revised manuscript, we repeated the CQ induced NF- κ B nuclear translocation in M2-like macrophages and presented a low-power view picture, indicating that 90-95% cells switch on NF- κ B.

According to the reviewer's suggestion, we added these results in the revised manuscript, page 11 line 251 and revised Figure 4f.

6. Figure 5l is lacking untreated and scrambled controls.

Response:

In the original Figure 5l, we actually used "+" and "-" to show CQ treatment and the untreated control, respectively, and used "Mcoln1-SGGFP" to show the scrambled control. In the revised manuscript, we remarked the Figure legend of Figure 5n to more clearly indicate untreated and scrambled controls.

7. How does 10 μ M CQ (the in vitro dose) correspond to 75 mg/kg (the clinical dose)?

Response:

In our *in vitro* study, we used 10 μ M CQ to treat macrophages, resulting in

effectively converting macrophages from M2 to M1 phenotype. However, when we increased the concentration to 20 μM , we found that some macrophages underwent apoptosis. Thus, we used 10 μM for *in vitro* experiments.

CQ has been widely studied before. It has been reported that 50mg/kg CQ administration results in 3-13 μM blood concentration (*Antimicrob Agents Chemother.*2011;55:3899-907; *Parasite.*1994;1:219-26.). Therefore, in our mouse model, we used 75mg/kg CQ to correspond to 10 μM *in vitro* concentration. In addition, in our study we found that 100mg/kg CQ caused obviously side effect in mice.

According to the reviewer's suggestion, we described this dosage in the Result section of the revised manuscript, page 5 line 104.

8. *Figure 2f and g: How was the CQ effect assessed here? The Clo treatment should have depleted the cells, so what was actually measured?*

Response:

In our original manuscript, we used nude mice to demonstrate that the antitumor effect of CQ is T cell dependent; then we used Clo-mediated depletion strategy to further demonstrate that the antitumor effect of CQ is macrophage dependent. Since CQ did not increase T cell function (see the original Supplementary Figure 2a) but increased macrophage function, we speculated that CQ indirectly enhanced T cell function via activating macrophages.

In original Figure 2f and g, we used CQ and Clo to treat B16 melanoma and H22 tumor ascites, respectively. We determined the function of tumor-infiltrating CD8⁺ T cells. We found that the expression of PD1 and CTLA-4 was downregulated in the CD8⁺ T cells after CQ treatment (see the revised Figure 2g,h).

According to the reviewer's comment, we rewrote the sentence of Figure 2g,h in the revised manuscript, page 7 line 159-162.

9. *The less acidic pH in lysosomes of M1 macrophages requires a reference.*

Response:

Lysosomal pH value is different between M1 and M2 macrophages. It has been reported that M2 macrophage lysosomes are more acidic than M1 lysosomes (*Mol Biol Cell.*2014;25: 3330-41.).

According to the reviewer's suggestion, we described the lysosomal pH feature of macrophages in the Result section of the revised manuscript, page 11 line 265, and added the reference in the revised manuscript as ref 34, page 33 line 858.

10. *Line 232, "however" makes no sense here.*

Response:

We deleted this word in revised manuscript.

11. Throughout the manuscript, some issues with grammar, word use and sentence structure.

Response:

We thank the reviewer's constructive suggestion. We carefully read the manuscript and corrected any grammar error, improved word use and sentence structure in the revised manuscript.

RESPONSE TO REVIEWER #2:

Reviewer #2 (Remarks to the Author):

The manuscript by Chen et al describes a role for lysosome pH in the regulation of M1 macrophage polarization. The authors show that some of the tumor suppressive effects of treating B16 melanoma or H22 hepatocellular carcinoma cells in vivo with the lysosome inhibitor, chloroquine is mediated by promoting a switch of tumor associated macrophages from the M2 to M1 tumor killing phenotype. CQ mediated M1-switching is shown to correlate with decreased MDSC and Treg infiltration and increased anti-tumor T cell function in vivo. Mechanistically, the authors show that decreasing lysosome pH following CQ treatment leads to MCOLN1 mediated Ca²⁺ release which correlates with increased p38 and NFκB signaling. In addition, the authors provide evidence that Ca²⁺ release is required for TFEB nuclear translocation and metabolic reprogramming of M1 switched macrophages. This study highlights a potentially important function of the lysosome in non-tumor cell populations. It also highlights the potential dual benefit of treating tumors with chloroquine given the potential for combined suppressive effects on the tumor cells as well as infiltrating macrophage populations. However clarification of some comments outlined below, would provide further clarity and novelty to this study.

-The authors only use CQ in all experiments. It would be valuable to show that another lysosome inhibitor – eg. Bafilomycin A1 – or knockdown of a V-ATPase subunit - also has the same effect.

Response:

We thank the reviewer's constructive suggestion. In the revised manuscript, we used bafilomycin A1 (Baf) to treat IL-4-conditioned M2 macrophages. We found that 10nM Baf less increased iNOS, IFN-γ, IL-12p40 expression, compared to CQ. And we increased Baf concentration to 50 nM, which however was toxic and induced apoptosis of macrophages. In addition, we knocked down Atp6v1c1 (a subunit of V-ATPase) in M2 macrophages. Also, the expression of iNOS, IFN-γ, IL-12p40 was weakly upregulated.

According to the reviewer's suggestion, we added these results in the revised manuscript, page 11 line 271 and revised Supplementary Figure 6b,c,d,e,f.

-TFEB is a known master regulator of lysosome biogenesis and function. Have the authors checked whether TFEB is responsible for the increase in lysosome biogenesis shown in Fig. 5c? Is there an increase in lamp1/2 staining in these cells?

Response:

In the revised manuscript, we knocked TFEB down in Raw264.7 macrophages and the lysosome numbers were analyzed by lysoTracker. The result showed that TFEB knockdown abrogated CQ-increased lysosome number. Moreover, we stained lamp1 and lamp2 in the macrophages. Consistently, CQ treatment increased lamp1 and lamp2 expression in macrophages, however TFEB knockdown disrupted this process.

According to the reviewer's suggestion, we added these results in the revised manuscript, page 12 line 281 and revised Figure 5d, page 12 line 289 and revised Supplementary Figure 6h,i,j.

-What are the relative expression levels of lysosome genes (or CLEAR element genes) in CQ treated macrophages?

Response:

Transcription factor EB (TFEB), a member of the basic helix-loop-helix leucine-zipper family of TFs, is a master regulator of lysosomal function. TFEB direct binds to the consensus sequence (TCACGTGA) at the promoters of lysosomal genes, and such TFEB-binding sequence is named as CLEAR (Coordinated Lysosomal Expression and Regulation) element (*Hum Mol Genet.*2011;20:3852-66.). In the revised manuscript, we found that CQ treatment upregulated the expression of CLEAR gene network (including gene *Arsa, Arsb, Atp6v0e1, Clcn7, Ctsa, Ctsb, Ctsd, Ctsf, Galns, Gba, Gla, Gns, Hexa, Lamp1, Lamp2, Naglu, Neu1, Psap, Scpep1, Sgsh, Tmem55b, Tpp1*) in IL-4-conditioned macrophages.

According to the reviewer's suggestion, we added these results in the revised manuscript, page 14 line 337 and revised Supplementary Figure 9e.

-Does TFEB KD impact lysosome gene expression, number, or pH?

Response:

In the revised manuscript, we knocked TFEB down in Raw264.7 cells, and found that TFEB knockdown inhibited lysosomal gene (*Lamp1 and Lamp2*) expression and decreased lysosome number in CQ-treated Raw264.7 macrophages. Intriguingly, TFEB knockdown did not affect CQ-increased pH value in macrophages.

According to the reviewer's suggestion, we added these results in the revised manuscript, page 12 line 289 and revised Supplementary Figure 6h,i,j,k.

-KD of TFEB was not able to promote switching from an M2 to M1 state (fig 6g). However KD of TFEB in CQ conditioned macrophages was sufficient to prevent a decrease in infiltrating MDSCs and Treg (fig 7j). How do the authors

reconcile these results?

Response:

In this study, we identified that CQ effects on lysosomes and increases the pH value of macrophages, leading to activating Ca^{2+} signaling. Such lysosome-triggered Ca^{2+} signaling activates p38 and NF- κ B for M1 phenotype on one hand, also activates TFEB for macrophage M1 metabolism on the other hand. If we block TFEB with siRNA, it only influences metabolic switch of macrophages but not their phenotype. Since metabolism is the basis for the function exertion, TFEB knockdown may impair macrophage function, leading to prevent CQ-decreased infiltration of MDSCs and Treg cells in tumor microenvironment. Thus, although TFEB has no effect on CQ-reset macrophage phenotype, TFEB knockdown seems to be sufficient to prevent CQ-decreased infiltration of MDSCs and Treg cells in tumor microenvironment via the macrophage-dependent pathway. This inconsistency might be reconciled by that TFEB-regulated glucose metabolism is required for the function exertion of CQ-reset macrophages

According to the reviewer's suggestion, we added this information in the Discussion section of the revised manuscript, page 20 line 488.

-Do the genes shown to be dependent on TFEB regulation (SLC2A1, SLC2A4, PFKM, PKLR) contain TFEB binding elements in their promoter (CLEAR elements)? Can the authors provide ChIP data to show direct regulation?

Response:

In our original manuscript, we used bioinformatical technology (JASPAR database) to predict the TFEB binding site, which indicates that there is consistent element that TFEB binds in the promoter of *Slc2a1*, *Slc2a4*, *Pfkm* and *Pklr*. TFEB knockdown resulted in the downregulation of these genes (see the original Figure 6f).

In the revised manuscript, we further conducted ChIP-qPCR assay. The result showed that TFEB indeed bound to the promoter of these genes.

According to the reviewer's suggestion, we added these results in the revised manuscript, page 14 line 335 and revised Supplementary Figure 9d.

-Nuclear localization is not obvious in panels 5e, l, n. Can the authors provide cytoplasmic vs nuclear fractionation via western blot to show a definitive change in cellular localization upon CQ treatment?

Response:

We appreciate the reviewer's constructive suggestion. In the revised manuscript, we extracted cytoplasmic and nuclear fraction, respectively, in CQ treated macrophages with or without CsA or in Mcoln1-KO Raw264.7 cells. Western blot analysis showed that CQ treatment effectively promoted the entry of TFEB and NF- κ B into the nucleus, however Mcoln1 knockout or CsA pretreatment abrogated the entry of NF- κ B into the nucleus.

According to the reviewer's suggestion, we added these results in the revised

manuscript, page 12 line 287 and revised Supplementary Figure 6g, page 13 line 308 and revised Supplementary Figure 7e,f.

-Level of IL-10 (M2) secretion in fig 3g could also be included.

Response:

We thank the reviewer's constructive suggestion. In the revised manuscript, we also analyzed the levels of IL-10, and the result showed that CQ treatment resulted in the decreased of IL-10 in tumor tissue.

According to the reviewer's suggestion, we added these results in the revised manuscript, page 9 line 214 and revised Figure 3i.

-some spelling and grammatical errors were found in the text.

Response:

We thank the reviewer for this constructive suggestion. We carefully read the manuscript and corrected any spelling and grammatical errors in the revised manuscript.

RESPONSE TO REVIEWER #3:

Reviewer #3 (Remarks to the Author):

The manuscript by Chen et al, describes a novel anti-tumoral action of chloroquine (CQ) that entails the switching of TAMs from M2 to a M1 tumor-killing phenotype supporting anti-tumor immunity responses, by reducing infiltration of immunosuppressive Tregs and MDSCs. The authors analyse the mechanisms of the effect of CQ on TAMs and find that it involves a Ca²⁺-mediated TFEB-dependent pathway reprogramming TAM metabolism. The study is relevant for our understanding of the in vivo effects of CQ. However, several aspects of this study remains to be further validated experimentally in order to support author's conclusions.

General remarks:

In all ex vivo or in vitro experiments with immune cells that the authors have presented, the authors have completely excluded the role of cancer cells. CQ can exert cytotoxic and immunosuppressive effects on or via cancer cells and these effects also play a role in immunological modulation. Considering that a tumor is usually >90% cancer cells, the authors cannot exclude the immunological effects of in their in vitro or in vivo experiments. Thus the immune cell co-culture experiments with DCs or macrophages presented in this manuscript should be repeated in presence of cancer cells as well (B16 and hepatocellular carcinoma) in order to rule in or rule out the contributory or inhibitory roles of cancer cells.

Response:

We thank the reviewer's insightful suggestion. In the revised manuscript, we

treated B16 or H22 with 10 μ M CQ for 24 hours, then the treated B16 or H22 cells were collected and co-cultured with macrophages or DCs at a 10:1 ratio (more tumor cell and less immune cells). We found that neither CQ-untreated nor CQ-treated tumor cells upregulated the expression of CD80, CD86, MHC II and CCR7, by contrast, these tumor cells enhanced arginase1 and IL-10 expression in macrophages. These data suggest that the CQ-effected tumor cells are not able to promote macrophage polarization to M1 phenotype. According to the reviewer's suggestion, we added these results in the revised manuscript, page 6 line 140 and revised Supplementary Figure 2d, page 9 line 204 and revised Supplementary Figure 4g.

The authors argue that CQ enhances tumor killing by resetting TAMs towards M1 but they don't provide any data on immune-mediated tumor cell killing. For macrophages, simply showing increase in iNOS levels does not mean that the macrophages are killing cancer cells via respiratory burst. This actually needs to be shown (cancer cell death in presence of CQ+macrophages or only macrophages and only CQ, plus, NO levels in media to start with) and iNOS or NO activity should be blocked to establish this. Also T cell-based cancer cell killing analysis should be performed. The authors moreover present no analysis on perforin-granzyme positivity of T cells in vivo, or whether T cells derived from CQ-treated tumors can kill B16 cells ex vivo/ in vitro. All these assays are necessary to validate conclusions of the authors (including the title of this paper).

Response:

We appreciate the reviewer's constructive suggestion. In the revised manuscript, we measured NO levels after CQ treatment and found that CQ treatment resulted in the increased NO levels in IL-4-conditioned M2 macrophages. We then co-cultured tumor cells with CQ-treated or untreated M2 macrophages. We found that CQ-treated M2 macrophages only slightly caused tumor cell death, and inhibition of NO production resulted in the disruption of such slight killing, suggesting that CQ-polarized macrophages may directly induce a small scale tumor cell death in a NO-dependent manner, especially considering that T cell depletion leads to abrogating CQ-mediated antitumor effect (see original Figure 1f,g,h,i). Then, we clarified the role of T cell immunity in CQ-mediated antitumor effect. OVA-B16 melanoma-bearing CD45.1⁺ C57BL/6 mice were treated with or without CQ, concomitant with adoptive transfer of OVA-specific CD8⁺ T cells from CD45.2⁺ OT-I transgenic mice. We found that CQ treatment enhanced tumor growth inhibition mediated by CD8⁺ T cells, which however was blocked by macrophage depletion. In line with this result, upon CQ treatment, tumor-infiltrating CD45.2⁺ T cells upregulated IFN- γ evidenced by flow cytometry as well as upregulated granzyme B in OVA-B16 tumor tissues evidenced by immunohistochemical staining. In addition, we isolated tumor-infiltrating CD45.2⁺ cells and co-cultured with OVA-B16 cells at a ratio of 30:1 for cytotoxicity assay. The

result showed that tumor-specific T cells from CQ-treated mice more efficiently killed OVA-B16 cells, compared to T cells from untreated mice. Together, these data suggest that CQ seems not to mobilize M1 macrophages polarized as the major killing arm against tumor but mobilizes polarized macrophages to remodel the antitumor microenvironment, thus facilitating the activation of tumor-specific T cells.

According to the reviewer's suggestion, we added these results in the revised manuscript, page 8 line 188 and revised Figure 3c, page 8 line 197 and revised Supplementary Figure 4f, page 7 line 167 and revised Figure 2i and Supplementary Figure 3d,e,f.

Specific points:

Figure 1. There is a discrepancy in the data comparing immunocompetent mice with immunodeficient ones. Why are immunocompetent mice always treated with higher dose of CQ (75mg/kg) whereas immunodeficient mice or nude mice are always treated with lower dose (50 mg/kg) of CQ? Similar doses of CQ should be used in both mice cohorts in order to rule out that differences in tumor growth are because of different doses of CQ rather than T cell dependency.

Response:

We thank the reviewer's constructive comments. In the revised manuscript, we used high dosage of CQ (75mg/kg) to treat nude mice bearing B16 melanoma or H22 hepatocarcinoma. The result was similar to that from 50 mg/kg treatment.

According to the reviewer's suggestion, we revised Figure 1f,g.

Figure 1. CD3 depletions assays need more statistical analysis to be valid. No statistics are shown for CQ+CD3 condition vs CQ alone. Is the effect of CD3 antibodies even significant? CQ only conditions has no error bars in some places? That is improbable.

Also no interpretation provided for why CD3 depletion has different effect on PBS and CQ conditions?

Response:

In the original manuscript, we used anti-mouse CD3 ϵ (clone 145-2C11) to deplete CD3⁺ T cells with a protocol of 200 μ g/mouse every 4 days via i.p. injection. T cell depletion with such protocol counteracted CQ-mediated antitumor effect (see original Fig. 1h). However, such protocol might not completely deplete CD3⁺ T cells, regarding to once depletion per 4 days. Notably, this antibody also has agonist effect that activates CD3⁺ T cells (*Immunity.2014;41:555-66; Immunity.2015;42:68-79.*). Such T cell activation might result in the inhibition of tumor growth (*Science.1988;242:569-71.*), as shown between the PBS group and CD3 depletion group in original Figure 1h. To verify this, in the revised manuscript, we additionally intraperitoneally injected 200 μ g anti-CD3 Ab to tumor-bearing mice once another day to better

deplete T cells. Under such condition, we found that T cell depletion accelerated tumor growth, compared to PBS group. Since the purpose of this figure was to further demonstrate T cell-dependent antitumor effect of CQ, we removed the PBS control from the original Figure 1h in order to make the result clearer and avoid any confusion. Also, we added all the statistics in the revised Figure 1h and i. In addition, we diminished the connecting line to better present the error bars.

According to the reviewer's suggestion, we revised Figure 1h and Supplementary Figure 1b.

The assay with splenic T cells that the authors carry out ex vivo to rule out direct effect of CQ on T cells is not representative of this effect. Splenic T cells should be derived from the immunocompetent mice treated in Fig 1 with CQ (75mg/kg) and these should be analyzed for markers of cell death (Annexin V-PI), activation (CD69, LAMP1, CD137) and T cell anergy (PD1, CTLA4).

Response:

We thank the reviewer's constructive suggestion. In the original manuscript, we isolated and treated splenic T cells with anti-CD3 and CD28 antibodies in the presence or absence of CQ. We did not observe the promoting effect of CQ on T cell proliferation. In the revised manuscript, we further isolated splenic T cells from CQ (75 mg/kg) treated and untreated mice, respectively. Consistently, T cells in CQ treated mice did not show difference from that in untreated mice, including T cell death (Annexin-V and PI staining), T cell activation (CD69, CD137 marker), lysosome status (Lamp1) as well as T cell anergy (PD1, CTLA-4).

According to the reviewer's suggestion, we added these results in the revised manuscript, page 6 line 130 and revised Supplementary Figure 2b.

Fig 2. The T cell immunophenotyping is not correctly done. CD8+IFN γ + cells include both CD8+T cells and NK cells - the authors need to quantify CD45+CD3+CD8+IFN γ + population for CD8+T cells. Moreover it is not clear what is the physiological relevance of results in Fig 2d,e. that is - 4% increase in CD8+IFN γ + cells is not really a huge amount for enforcing CD8+T cell-driven immunity. The authors should better deplete CD8+ cells with antibodies to confirm the relevance of this population in this system. Also it is not clear what population is labelled with CTLA4 or PD1? More markers should be used for T cells, macrophages/myeloid cells and cancer cells to differentiate between different tumoral populations expressing these immune-checkpoints. The correlation between IFN γ and PD1/CTLA4 levels makes no sense. Sustained IFN γ production up-regulates PD1/CTLA4 as a negative feedback loop mechanism, this is the whole basis of the concept of immune-checkpoints, whereas the authors see the opposite of this. This data needs to be clarified better. More later time-points should be added and direct effects of CQ on PD1/CTLA4 should be better presented.

Response:

CD8 molecule recognizes and binds MHC class I α chain, which is expressed in CD8⁺ T cells and also expressed in some DCs and NK cells. In our original manuscript, we actually gated the CD3⁺ cells and then analyzed CD8⁺IFN- γ ⁺ cells (please see the figure legend of Figure 2d,e in the original manuscript, page 33 line 813-819). For Figure 2d, e, we isolated immune cells from tumor tissues and directly analyzed CD3-gated CD8⁺IFN- γ ⁺ cells by flow cytometry without activation of these cells *in vitro* as well as addition of Golgi inhibitor. In this case, IFN- γ is able to be released to the extracellular space and the intracellular detection of IFN- γ is low.

In the revised manuscript, we isolated the tumor-infiltrating immune cells and treated them with anti-CD3 and CD28 antibodies *in vitro* for 24 hours. Monensin was added for the last 6 hours to block Golgi transport function. Upon this *in vitro* stimulation, we found that more than 20% CD3⁺CD8⁺ T cells are IFN- γ positive. We used this higher percentage of IFN- γ ⁺ in CD3⁺CD8⁺ T cells to replace the original one (original Figure 2d,e). We also conducted CD8 T cell depletion experiment to further demonstrate that CQ treatment causes IFN- γ production by CD8 T cells. We found that CD8 T cell depletion with anti-CD8 depleting antibody resulted in the loss of most IFN- γ in tumor tissue. According to the reviewer's suggestion, we revised Figure 2d,e and added new results in the revised manuscript, page 7 line 156 and revised Figure 2f.

PD-1 and CTLA-4 are two important inhibitory molecules expressed on T cell membrane surface. Although IFN- γ is capable of upregulating PD-L1 and IDO1 to induce negative immunoregulation, it does not regulate PD-1 expression. To date, the mechanism of PD-1 upregulation in tumor-infiltrating T cells remains unclear. On the other hand, IFN- γ generally does not regulate CTLA-4 expression. CTLA-4 is commonly expressed on T cell membrane, which however is internalized upon tyrosine phosphorylation of cytoplasmic tail (*J Cell Biochem.2000;78:241-50.*) and can be recycled to the membrane or undergo degradation upon signals T cells receive (*J Biol Chem.2012;287:9429-40.*). In this study, we observed that CQ treatment resulted in the upregulation of IFN- γ and downregulation of PD1/CTLA-4 in tumor-infiltrating CD8⁺ T cells. To further verify this, we additionally analyzed T-bet, an essential transcription factor for IFN- γ production in CD8⁺ T cells (*Science.2002;295:338-42; Proc. Natl. Acad. Sci. USA.2002;99:5545-50.*). We found that T-bet was upregulated in tumor-infiltrating CD3⁺CD8⁺ T cells upon CQ treatment, while the expression of PD1/CTLA-4 was downregulated in CD3⁺CD8⁺T-bet⁺ T cells, consistent with previous report that T-bet represses PD-1 expression (*Nat Immunol.2011;12:663-71.*).

According to the reviewer's suggestion, we added these results in the revised manuscript, page 7 line 162 and revised Supplementary Figure 3c.

IL12p40 analysis is not a correct marker for macrophage activation. IL12p70 is the actual active subunit of IL12. While IL12p40 is generally easily

up-regulated, in absence of IL12p70, IL12p40 cannot enforce any immunological activity. Analysis for IL12p70 should also be provided and to support author's conclusions, it should follow the same trend as IL12p40.

Response:

We thank the reviewer's constructive suggestion. In the revised manuscript, we measured IL-12p70 in CQ-treated macrophages both mouse and human by ELISA. In line with IL-12p40, IL-12p70 production was elevated upon CQ treatment.

According to the reviewer's suggestion, we added these results in the revised manuscript, page 8 line 189 and revised Supplementary Figure 4c, page 9 line 222 and revised Supplementary Figure 4j.

The authors have analyzed the level of immunostimulatory M1 cytokines but not those of immunosuppressive M2 cytokines. IL10, TGF β and IDO1 levels should also be analyzed in media of M1 and M2 assays to clarify functional M2 polarisations and address the possibilities of partial functional suppression within M1 macrophages.

Response:

In the revised manuscript, we additionally analyzed IL-10, TGF- β 1 and IDO1 in CQ-treated M2 macrophages by ELISA (IL-10, TGF- β 1) or real-time PCR (IL-10, TGF- β 1, IDO1). We found that the expression of IL-10, TGF- β 1 and IDO1 was downregulated with CQ treatment.

According to the reviewer's suggestion, we added these results in the revised manuscript, page 8 line 195 and revised Supplementary Figure 4e.

Fig 3. 10-15% positivity of total macrophages for respective cytokines isn't very physiologically high. The authors need to show actual concentrations of these cytokines in the medium and these concentrations need to (significantly) exceed 10 pg/mL to be physiologically relevant.

Response:

In the revised manuscript, we detected the concentration of IFN- γ , IL-12p70 and TNF- α by ELISA. The result showed that CQ treatment elevated these cytokines' concentration to 100-800pg/mL.

According to the reviewer's suggestion, we added these results in the revised manuscript, page 8 line 189 and revised Supplementary Figure 4c.

Fig 3. Why are macrophages in Fig 3b already more than 60% positive for CD80? This means that the macrophages are pre-stimulated on a certain level. The authors need to optimize the macrophage cultures to have CD80 positivity below 20%.

Response:

In the original manuscript, we pre-incubate macrophages on ice with anti-CD16/CD32 (clone 93 mAb) to block non-specific binding of immunoglobulin to macrophage Fc receptors. We then conducted flow

cytometric analysis. The high proportion of CD80 might be due to that the CD16/CD32 Ab lost its efficacy. In the revised manuscript, we used a new one and repeated the assay again. The result showed a consistent trend but the CD80 positive proportion became 15%.

According to the reviewer's suggestion, we revised Figure 3b,d.

Fig 7. CD11b+Gr1(hi) are not a specific markers for MDSCs. These markers mark all neutrophils and granulocytes. In order to have a valid MDSC panel the authors need to have CD11b+GR1(hi)CD86(lo)MHCII(lo) wherein low expression of CD86 and MHCII is marker of the possible suppressive nature of these granulocytes. Also MDSCs, are not just granulocytic in origin but also myeloid-MDSCs need to be analyzed by a similar strategy. The authors need to deplete MDSCs in vivo to prove their conclusions regarding MDSCs.

Response:

Accumulation of immature myeloid cells and immune suppression was recognized in both tumor-bearing mice and cancer patients in 1980s (*Cancer Res.1987;1:100-5; J Immunol.1989;2:491-8.*). Because of the suppressive activities and heterogeneity of these myeloid cells, the term myeloid-derived suppressor cell (MDSC) was proposed to denote this population (*Cancer Res.2007;1:425.*). MDSCs potentially differentiate toward monocyte/macrophages or granulocytes. Correspondingly, MDSCs in mouse can be classified as monocytic MDSCs (CD11b⁺Ly6C^{hi}Ly6G⁻) and granulocytic MDSCs (CD11b⁺Ly6C^{lo}Ly6G⁺) and both of them favor tumor growth in many tumor models (*Cancer Cell.2017;32:654-68; Cancer Discov.2017;1:72-85; Cancer Cell.2016;30:108-19; Trends Immunol.2016;37:208-20.*). CD11b⁺Gr-1⁺ MDSCs actually contain these two subsets, since the anti-Gr-1 antibody (RB6-8C5) recognizes both Ly6C and Ly6G. Thus, in the original manuscript, we did not further clarify which subsets of MDSCs in mouse tumor model was influenced by CQ treatment. In the revised manuscript, we further analyzed CD11b⁺Ly6C^{hi}Ly6G⁻ monocytic and CD11b⁺Ly6C^{lo}Ly6G⁺ granulocytic MDSCs. We found that CQ treatment actually decreased both subsets of MDSCs. Such CQ-mediated decrease could be disrupted by macrophage depletion, Mcoln1 Knockout or TFEB knockdown.

According to the reviewer's suggestion, we added these results in the revised manuscript, page 15 line 366 and revised Supplementary Figure 10a,b,c,d,f.

We previously did a lot of work on MDSCs and used Gr-1 antibody to efficiently deplete MDSCs (*Cancer Res.2006;66:1123-31; Cancer Lett.2007;252:86-92; PLoS One.2010;5:e8922; J Immunol.2010;185:7199-206.*). In the revised manuscript, we further used anti-Gr-1 depleting antibody (BioLegend) to deplete both granulocytic and monocytic MDSCs. Under this condition, CQ-mediated antitumor effect was impaired in B16 melanoma tumor model.

According to the reviewer's suggestion, we added these results in the revised manuscript, page 16 line 386 and revised Supplementary Figure 10j.

Cyclophosphamide is not a selective Treg depleting methodology, it has several other effects on the tumor (e.g. immunogenic cell death) and gut microbiota (Th17 anti-tumor axes). These data should be replaced by Tregs depleted in the current system via anti-CD25 antibodies. Finally chlodronate liposome-based depletion is not specific for macrophages only but targets all phagocytes.

Response:

Cyclophosphamide (CY) is a prominent alkylating anticancer agent that induces tumor cells into death. Low-dose CY, although not directly killing tumor cells, is able to activate antitumor immunity by selective depletion of Tregs, leading to therapeutic effect against tumors. We previously elucidated the underlying mechanism through which low-dose CY selectively deplete Treg cells but not conventional CD4 T cells (*Cancer Res.*2010;70:4850-8.); we further found that 25 mg/kg dosage of CY is optimal for Treg depletion in mouse tumor model (*J Immunol.*2010;185:7199-206.). With high dosage (such as 100 mg/kg) of CY treatment, tumor cells can be killed and release tumor antigens and danger signals, leading to immunogenic cell death (*Cancer Cell.*2015;28:690-714.). Also, under high dosage (100 mg/kg) condition, CY destroyed gut integrity, leading to Th17 antitumor axes (*Science.*2013;342:971-6.). However, in this study, we used low-dose CY (25 mg/kg), which did not result in immunogenic cell death nor activated Th17 antitumor axes. In the revised manuscript, we additionally used anti-CD25 depleting antibody (PC-61), a widely used method to deplete Treg cells in mouse model (*Nature.*2016;530:434-40; *Nature Commun.*2015;6:7566; *Leukemia.*2015;29:947-57.). Consistent with low dose CY effect, CD25 depletion also enhanced the inhibitory effect of CQ on B16 or H22 tumor growth.

According to the reviewer's suggestion, we added these results in the revised manuscript, page 16 line 384 and revised Supplementary Figure 10h,i.

Macrophages are critical innate immune cell subset with high heterogeneity. The completely specific marker for macrophages is not identified. There always are cross-marker(s) and cross-function(s) between macrophages and other immune cells. Currently, there is no specific agent to target macrophages. Chlodronate liposome since its discovery is widely used to deplete macrophages in various tumor models as a relatively specific agent, and their feasibility and efficiency have also been well demonstrated (*Nat Commun.*2016;7:11051; *Cancer Cell.*2014;26:190-206; *Nature Medicine.*2013;19:1166-72; *Nature Medicine.*2013;19:429-36; *Nature Immunology.*2013;14:574-83.).

REVIEWERS' COMMENTS:

Reviewer #1 (Remarks to the Author):

All technical and statistical concerns have been addressed.

Reviewer #2 (Remarks to the Author):

The authors have sufficiently addressed my questions and recommendations for improving the manuscript. A significant amount of additional data has been added that further strengthen the claims of the study. Overall the findings by Chen, Xie et al will be broadly applicable to the community and warrant acceptance for publication in Nature Communications.

Reviewer #3 (Remarks to the Author):

I am happy to see that the authors have taken into serious consideration most of the comments raised by this reviewer and performed a battery of new experiments supporting their conclusions. The manuscript has been largely improved and several inconsistencies have been solved.

I would nevertheless pinpoint that some of the additional sentences need to be more carefully checked for grammatical mistakes.